# A DHODH inhibitor increases p53 synthesis and enhances tumor cell killing by p53 degradation blockage

Marcus J.G.W. Ladds, Ingeborg M.M. van Leeuwen, Catherine J. Drummond et al.[#]

The development of non-genotoxic therapies that activate wild-type p53 in tumors is of great interest since the discovery of p53 as a tumor suppressor. Here we report the identification of over 100 small-molecules activating p53 in cells. We elucidate the mechanism of action of a chiral tetrahydroindazole (HZ00), and through target deconvolution, we deduce that its active enantiomer (*R*)-HZ00, inhibits dihydroorotate dehydrogenase (DHODH). The chiral specificity of HZ05, a more potent analog, is revealed by the crystal structure of the (*R*)-HZ05/DHODH complex. Twelve other DHODH inhibitor chemotypes are detailed among the p53 activators, which identifies DHODH as a frequent target for structurally diverse compounds. We observe that HZ compounds accumulate cancer cells in S-phase, increase p53 synthesis, and synergize with an inhibitor of p53 degradation to reduce tumor growth in vivo. We, therefore, propose a strategy to promote cancer cell killing by p53 instead of its reversible cell cycle arresting effect.

Targeted therapeutics have demonstrated limited clinical success and frequently need to be used in combination to improve efficacy. For example, while imatinib and second-generation ABL tyrosine-kinase inhibitors can hold chronic myeloid leukemia (CML) in remission, new strategies to eliminate CML stem cells are needed to achieve cure[1]. As previously shown, p53 activation can enhance elimination of CML stem cells in combination with imatinib[2]. This suggests the potential therapeutic utility of p53 activating agents in combinatorial studies with other targeted therapeutics to enhance their efficacy.

A number of small molecules and peptides have been identified that impair the interaction of p53 with mdm2, an important negative regulator of p53 stability[3]. Although nutlin-3[4], the most commonly used mdm2 inhibitor, has cytotoxic effects in some cell types, it frequently induces a reversible cell cycle arrest that could limit its efficacy in cancer treatment. Due to this, nutlin-3 can even protect cancer cells from standard chemotherapy that relies upon cells progressing through the cell cycle[5]. Moreover, mdm2 inhibitors may exhibit on-target clinical toxicity[6] and can predispose cells to genomic instability[7]. Therefore, if p53 activation is to be exploited as a cancer therapy, it is critical to find compound combinations that reduce the dose of mdm2 inhibitors required, while also enhancing their ability to kill cells rather than inducing their reversible cell cycle arresting effects.

We identify, from our phenotypic screen of activators of wild-type p53 transcriptional function, that dihydroorotate dehydrogenase (DHODH) is a remarkably frequent target for activators of p53. Additionally, we examine a chiral tetra-hydroindazole (HZ00) and its more potent analog as inhibitors of DHODH following target deconvolution. Through our work, we uncover the therapeutic implications of DHODH inhibition, which include the accumulation of cells in S-phase, an increase in p53 synthesis, and enhancement of the antitumor effect of an inhibitor of p53 degradation.

## Results

**Identification and characterization of HZ00.** A phenotypic screen was performed to identify activators of p53-dependent transcription. A total of 20,000 small molecules were tested using two p53 wild-type reporter cell lines, a human melanoma cell line (ARN8) and a murine fibroblast cell line (T22) (Fig. 1a). Twenty compounds were shown to activate p53 in ARN8 melanoma cells >1.5-fold and did not activate or did so below 1.5-fold in T22 fibroblasts. We focused on one of these 20 cell selective molecules and named this tetrahydroindazole HZ00 (1) (Fig. 1b). Figure 1c shows the increase in p53-reporter activity by HZ00 in ARN8 cells. HZ00 did not result in p53-reporter activity in p53-null H1299 cells (Supplementary Fig. 1a). HZ00 also raised the mRNA of p53 target genes in ARN8 cells (Supplementary Fig. 1b and Supplementary Data 1).

HZ00 increased the levels of p53 and p53-induced proteins in ARN8 melanoma cells (Fig. 1d). The rise in protein levels of p53 and its downstream targets in response to HZ00 were weaker in human normal dermal fibroblasts (HNDFs) (Fig. 1d) and did not occur in p53-deficient cells (Supplementary Fig. 1c). Accordingly, HZ00 had a lesser inhibitory effect on proliferation of HNDFs than on the growth of ARN8 cells (Fig. 1e).

HZ00 also differs from inhibitor of p53 degradation, nutlin-3, as it does not bind to human mdm2 (hdm2) (Fig. 1f and Supplementary Fig. 2a). Binding to human mdmx (hdmx), another negative modulator of p53, was also not detected (Fig. 1f and Supplementary Fig. 2a) nor were there indications of p53 stabilization according to experiments performed in the presence of cycloheximide (Supplementary Fig. 2b). In addition, we did not observe any induction of DNA damage response markers (Supplementary Figs. 2c−h). Instead, HZ00 was able to increase p53 synthesis within 6 h of treatment (Fig. 1g). When cells were pulse labeled with $^{35}$S methionine/cysteine for 30 min we detected a robust increase in the amount of newly synthesized p53 in contrast to the results seen with the inhibitor of p53 degradation, nutlin-3. This increase in p53 synthesis by HZ00 coincides with the time at which p53 protein levels start to rise (Supplementary Fig. 2f) but is not accompanied by a commensurate rise in p53 mRNA (Supplementary Fig. 1d and ENSG00000141510 in Supplementary Data 1).

One important feature of HZ00 is that it is not a pan assay interference compound (PAIN)[8]. In support of a non-promiscuous profile, we did not detect inhibition by HZ00 of any of the wide panel of kinases that we tested (Supplementary Tables 1−3). In addition, HZ00 possesses a stereogenic center. We describe a synthetic route for the HZ00 racemic mix (for simplicity, the name HZ00 is used in this text to refer to the racemic mixture) and each of its enantiomers (Fig. 2a). As shown in Fig. 2b, the (R)-enantiomer of HZ00 led to significantly higher p53 reporter activity in ARN8 cultures than its (S)-enantiomer suggesting that (R)-HZ00 could be selective for one or few cellular targets.

HZ00 was not only more selective for ARN8 cancer cells than HNDF normal cells when compared to nutlin-3 in MTT assays (Fig. 1e) but, unlike nutlin-3, it also caused a sharp increase in the sub-G1 population in ARN8 cells (Fig. 2c). We also noticed that combining HZ00 with nutlin-3 increased the percentage of sub-G1 cells even further. In contrast, in HNDF cultures, HZ00 did not reduce proliferation after 48 h nor did it substantially enhance cell death in the presence of nutlin-3 (Fig. 2c). We also tested whether the combination of HZ00 with nutlin-3 was synergistic. Indeed, according to two different models for synergy, HZ00 and nutlin-3 exhibited a synergistic cell kill on ARN8 cells in vitro (Fig. 2d).

These findings, together with the favorable in vitro pharmacokinetic properties (Supplementary Table 4) encouraged us to investigate (R)-HZ00 in vivo in combination with nutlin-3. A significant ARN8 xenograft tumor growth inhibition was observed in response to the (R)-HZ00 and nutlin-3 combination (Fig. 2e). Although (R)-HZ00 was not toxic, treatments had to be discontinued after 9 days due to the toxicity of nutlin-3.

**(R)-HZ00 is a DHODH inhibitor.** The results described above persuaded us to elucidate the mechanism of action of HZ00. By performing a time course analysis using BrdU/PI flow cytometry, we observed that a short HZ00 treatment accumulates ARN8 cells in S-phase (Fig. 3a). At 49 h cytotoxicity was observed in a large proportion of ARN8 cells (Fig. 3b). However, unlike the deoxynucleotide synthesis inhibitor hydroxyurea, HZ00 did not increase levels of markers for replication fork stalling (Supplementary Figs. 2c and h) and instead reduced expression of cdc6, an ATPase involved in the licensing of replication origins (Supplementary Figs. 2i and j). Interestingly, cdc6 is repressed by the p53-DREAM complex[9] and its repression appears to be p21 dependent (Supplementary Fig. 2k). S-phase accumulation in response to HZ00 also occurred in other cell lines (Supplementary Fig. 3). In contrast, there were negligible changes in the cell cycle profile of HNDF cultures at the 49 h time point (Fig. 3b).

Additionally, we observed that nucleoli were disrupted early upon HZ00 treatment (Supplementary Fig. 4a). A drop in total RNA levels in ARN8 cells treated with HZ00 was also evident (Supplementary Fig. 4b). Following these observations, we asked whether addition of nucleosides could alter the cellular response

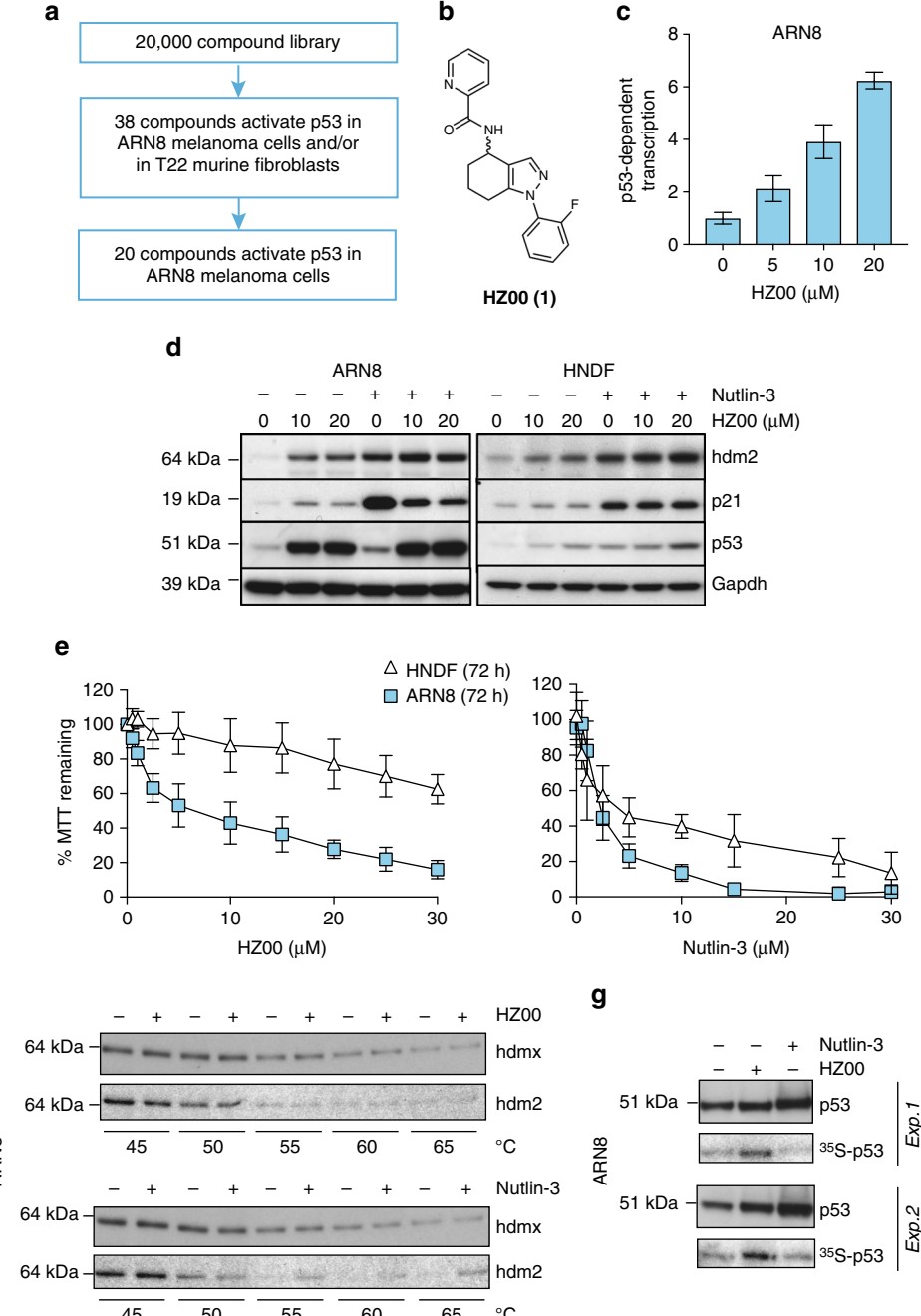

**Fig. 1** Discovery and activity of HZ00. **a** Compound library screening strategy to identify 20 compounds capable of activating p53 transcriptional function in ARN8 melanoma cells >1.5-fold and did not activate or did so below 1.5-fold in T22 fibroblasts. **b** The structure of HZ00 (**1**). **c** p53 wild-type ARN8 human melanoma cells were treated for 16 h with HZ00 and the level of p53-dependent transcription measured by CPRG assay. Values correspond to the average of three technical repeats ± SD and are representative of 5 biological replicates. **d** ARN8 or HNDF were treated with HZ00 for 1 h and then 2 μM nutlin-3 was added for an additional 18 h. Levels of p53, as well as downstream targets hdm2 (human mdm2) and p21 were determined. Levels of gapdh were used to monitor protein loading. **e** ARN8 or HNDF cells were treated with the indicated compound concentrations for 72 h and subjected to MTT assays. Values correspond to the average of 4 (HZ00) or 3 (nutlin-3) biological replicates ± SD. **f** ARN8 soluble cell extracts were prepared for the cellular thermal shift assay (CETSA) in PBS as described[37] and subjected to increasing temperatures in the absence or presence of 100 μM HZ00 or nutlin-3. Samples were centrifuged and hdm2 or hdmx were detected in the supernatants. **g** ARN8 cells were treated with 20 μM HZ00, 5 μM nutlin-3 or vehicle (DMSO) for 5 h 50 min and pulse labeled with $^{35}$S-Met-Cys for 30 min (6 h 20 min total). p53 was immunoprecipitated and p53 protein levels were determined by western blotting. Incorporation of $^{35}$S in the p53 immunoprecipitate was determined by autoradiography. The experiment is shown in duplicate

to HZ00. As shown in Fig. 4a, b, HZ00's effect on ARN8 cell growth and p53 activation was completely abolished by supplementation with high concentrations of uridine, suggesting inhibition of an enzyme involved in the de novo synthesis of UMP (Fig. 4c). The ablation of HZ00 activity was not achieved by adding other nucleosides. Next, we examined whether two intermediary metabolites in the de novo synthesis of UMP protected cells from HZ00 treatment. As shown in Fig. 4d, orotic acid but not dihydroorotic acid rescued ARN8 cells from (R)-HZ00. Consistent with our results using orotic acid and

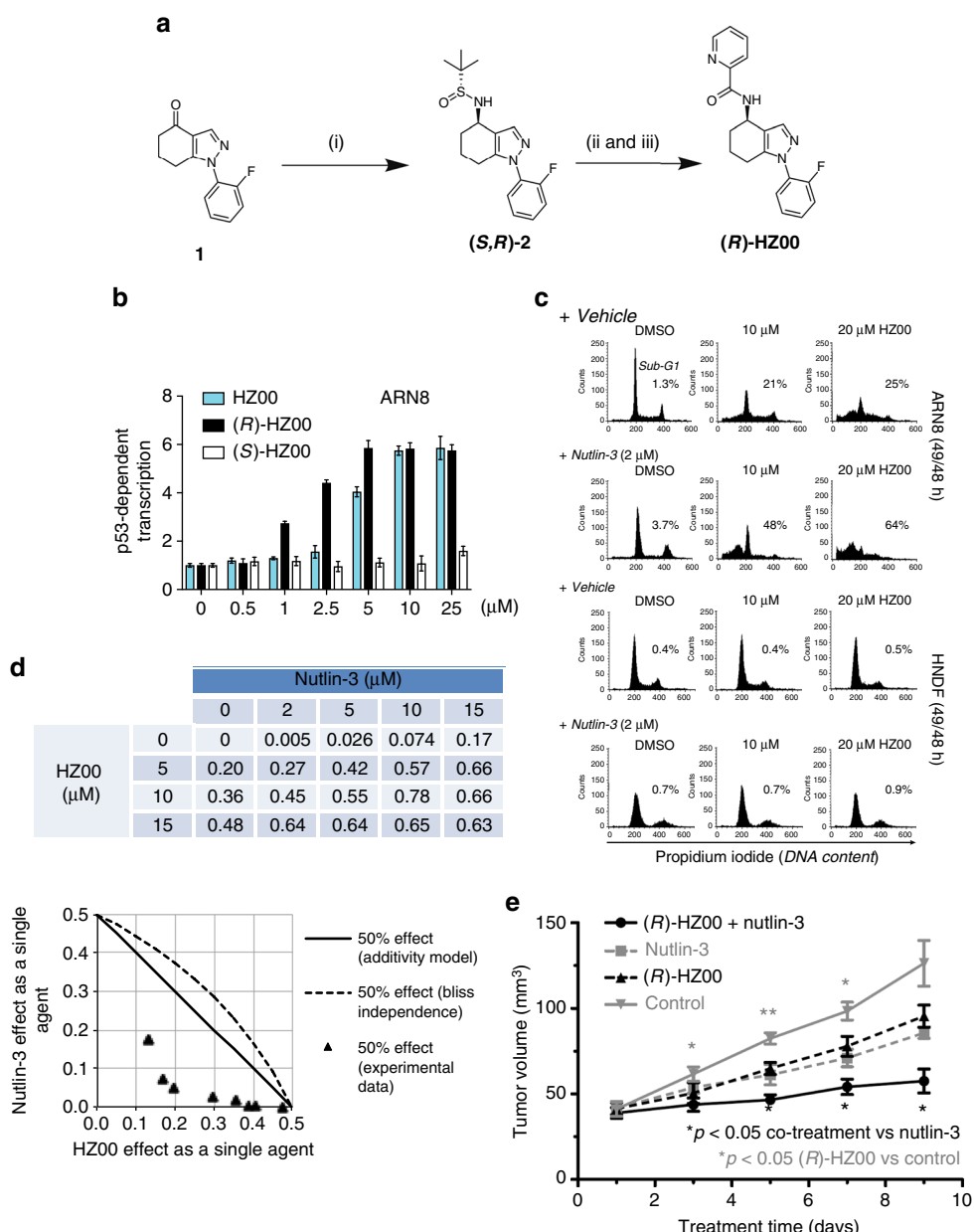

**Fig. 2** Synthesis and characterization of HZ00 and its enantiomers. **a** Synthetic route to (R)-HZ00 involving the use of Ellman's chiral auxiliary. Reagents and conditions: (i) (1.1) ($S_S$)-2-methyl-2-propanesulfinamide, Ti(OEt)$_4$, THF, 75 °C, (1.2) L-Selectride, THF, −48 °C, 73%, diastereomeric excess (d.e. 84 %). (ii) HCl, MeOH, RT; (iii) 2-picolinic acid, EDC.HCl, Et$_3$N, HOBt, DMAP, DCM, RT, 82% over 2 steps. **b** p53 wild-type ARN8 human melanoma cells were treated for 16 h with the indicated compounds and the level of p53-dependent transcription measured by CPRG assay. Values correspond to the average of 3 technical repeats ± SD. **c** ARN8 or HNDF cells were treated with HZ00 for 1 h and then 2 μM nutlin-3 was added for an additional 48 h. Cells were fixed and stained with propidium iodide (PI) and analyzed by flow cytometry. The percentages of sub-G1 cells are indicated. **d** ARN8 cells were treated for 72 h with HZ00 and/or nutlin-3 at the indicated doses. After treatment, cell cycle distribution was analyzed by flow cytometry following staining with PI. The effect of the compounds was quantified by obtaining the percentage of cells in sub-G1. The table shows the DMSO control subtracted effect for each dose combination ($d_H$, $d_N$). The curves in the normalized EC$_{50}$ isobologram for the HZ00-nutlin-3 combination indicate single-effect pairs ($x$, $y$) = (Eff[$d_H$]/100, Eff[$d_N$]/100) expected to give a 0.5 effect in combination according to the additivity (solid line) and Bliss independence (dashed line) models[38], respectively. Data points (triangles) indicate pairs ($x$, $y$) that give 0.5 effect based on linear interpolation of the experimental data shown in the combination matrix above. **e** The combination of (R)-HZ00 (150 mg kg$^{-1}$) and nutlin-3 (100 mg kg$^{-1}$) was assayed in a xenograft model of ARN8, and significantly inhibited growth in this model compared with control groups (i.e., (R)-HZ00, nutlin-3 or vehicle). $n = 5$ mice per group. Error bars illustrate ± SEM. (***$p < 0.001$, **$p < 0.01$ and *$p < 0.05$). P-values were calculated using multiple Student's t-tests

dihydroorotate supplementation, known DHODH inhibitors brequinar and teriflunomide (A77 1726)[10] phenocopied the effects of HZ00 (Fig. 4e). These observations indicated that HZ00 primarily targets DHODH. As uridine is also an important component in blood, we tested whether HZ00 was still able to reduce cell growth of tumor cells at physiological levels of uridine using literature values which report human plasma uridine concentration as between 2.5 and 4.9 μM and murine plasma uridine concentrations from 1.2 to 3.2 μM[11]. Indeed, we found that (R)-HZ00 was still successfully able to markedly reduce tumor cell growth at 2.5 μM uridine and even at up to 5 μM uridine (Fig. 4f).

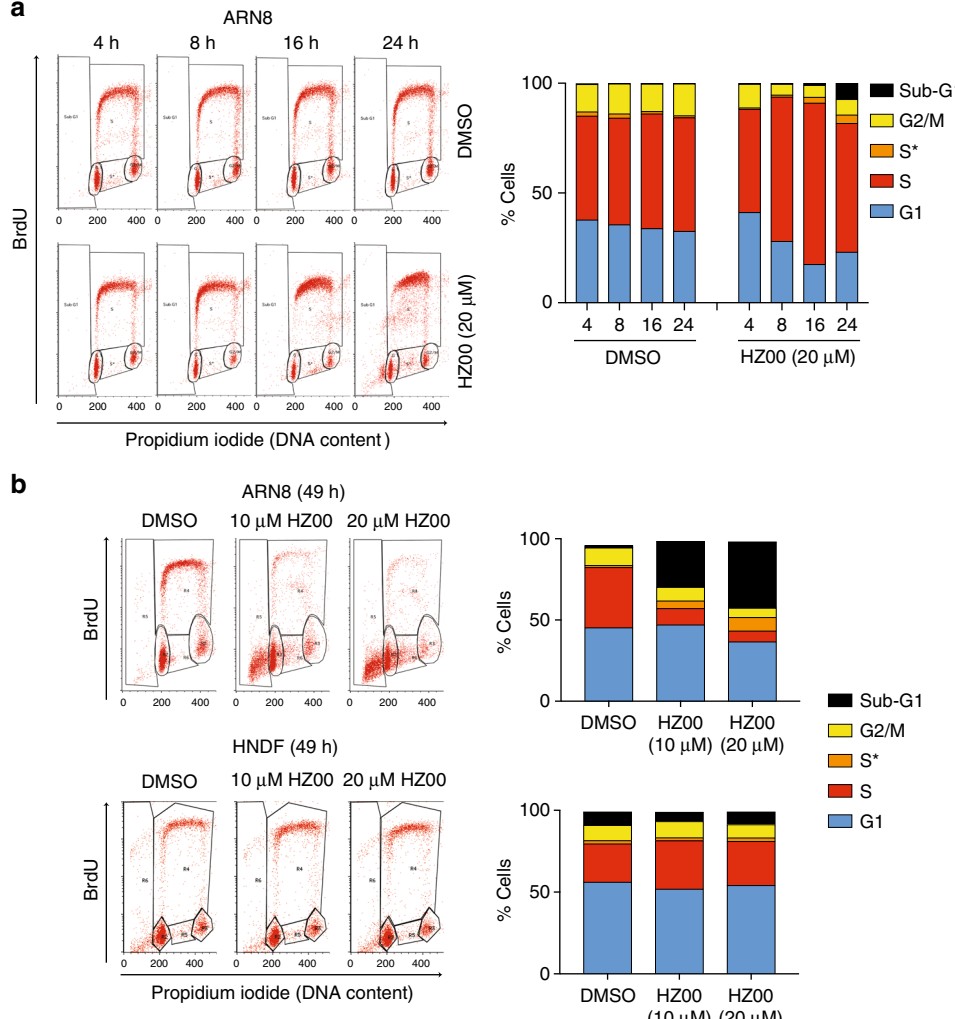

**Fig. 3** HZ00 accumulates cancer cells in S-phase. **a**, **b** ARN8 or HNDF cells were treated with HZ00 for the indicated times and analyzed by BrdU/PI flow cytometry with the percentage in each cell cycle phase graphed. S* indicates cells with a DNA content between 2N and 4N that do not incorporate BrdU

**p53 activators frequently inhibit DHODH**. An enzymatic assay using purified human DHODH confirmed that HZ00 is indeed a DHODH inhibitor and that (R)-HZ00 is significantly more potent than (S)-HZ00. In these assays, we used coenzyme $Q_{10}$ as an electron acceptor and the following $IC_{50}$ values were obtained: HZ00 (2.2 μM), (R)-HZ00 (1.0 μM), and (S)-HZ00 (9.5 μM).

We then tested whether any of the other p53 activating compounds in the 20,000 compound library inhibited human DHODH. Furthermore, we re-screened another 30,000 compounds, which had previously been tested in T22-RGCΔFos-LacZ murine fibroblasts[12], for their ability to activate p53 in ARN8 cells. In both screens, we found that a high proportion of compounds that activated p53 in ARN8 cells also inhibited DHODH (Supplementary Tables 5−6).

In summary, we present a large series of small molecules, in addition to the HZ series, that enhance p53-reporter activity (Supplementary Tables 5−7). These cluster into 31 chemotypes. Of these 31 chemotypes, 12 include compounds that inhibit DHODH by >40% at 10 μM.

**Testing of HZ analogs reveals HZ05 as a potent DHODH inhibitor**. After conducting the initial screen for DHODH inhibitors, we improved the DHODH activity assay by using 3,4-dimethoxy-5-methyl-p-benzoquinone (a more water soluble alternative to coenzyme $Q_{10}$) and by using a kinetic assay rather than an endpoint assay for enzyme activity. We then initiated a structure activity relationship (SAR) study by testing commercially available analogs of HZ00. Of the 29 HZ racemic mixtures tested, only two (HZ02 (**2**) and HZ05 (**3**)) were able to inhibit DHODH at nanomolar concentrations. HZ05 (Fig. 5a) was the most potent among these analogs and markedly more potent than HZ00.

To investigate the binding mode of the tetrahydroindazole series, we carried out crystallographic studies with HZ05 (racemic mix) and human DHODH (Fig. 5b, c). This resulted in a co-crystal structure of the (R)-enantiomer of HZ05 with DHODH at 1.7 Å resolution (PDB code 6ET4). The preference for the (R)-HZ05 enantiomer is consistent with the superior inhibitory potency of (R)-HZ05 versus (S)-HZ05 against DHODH (see $IC_{50}$ values in Fig. 5a). Like many other DHODH inhibitors, including brequinar and teriflunomide, (R)-HZ05 binds to the region referred to as the quinone tunnel[13]. However, the interactions of (R)-HZ05 with the protein are different to brequinar and related inhibitors. When (R)-HZ05 is bound to DHODH, Gln47 is displaced from the pocket allowing interactions between (R)-HZ05 and Arg136 through water molecules. The co-crystal structure also provides an explanation for the poor inhibitory

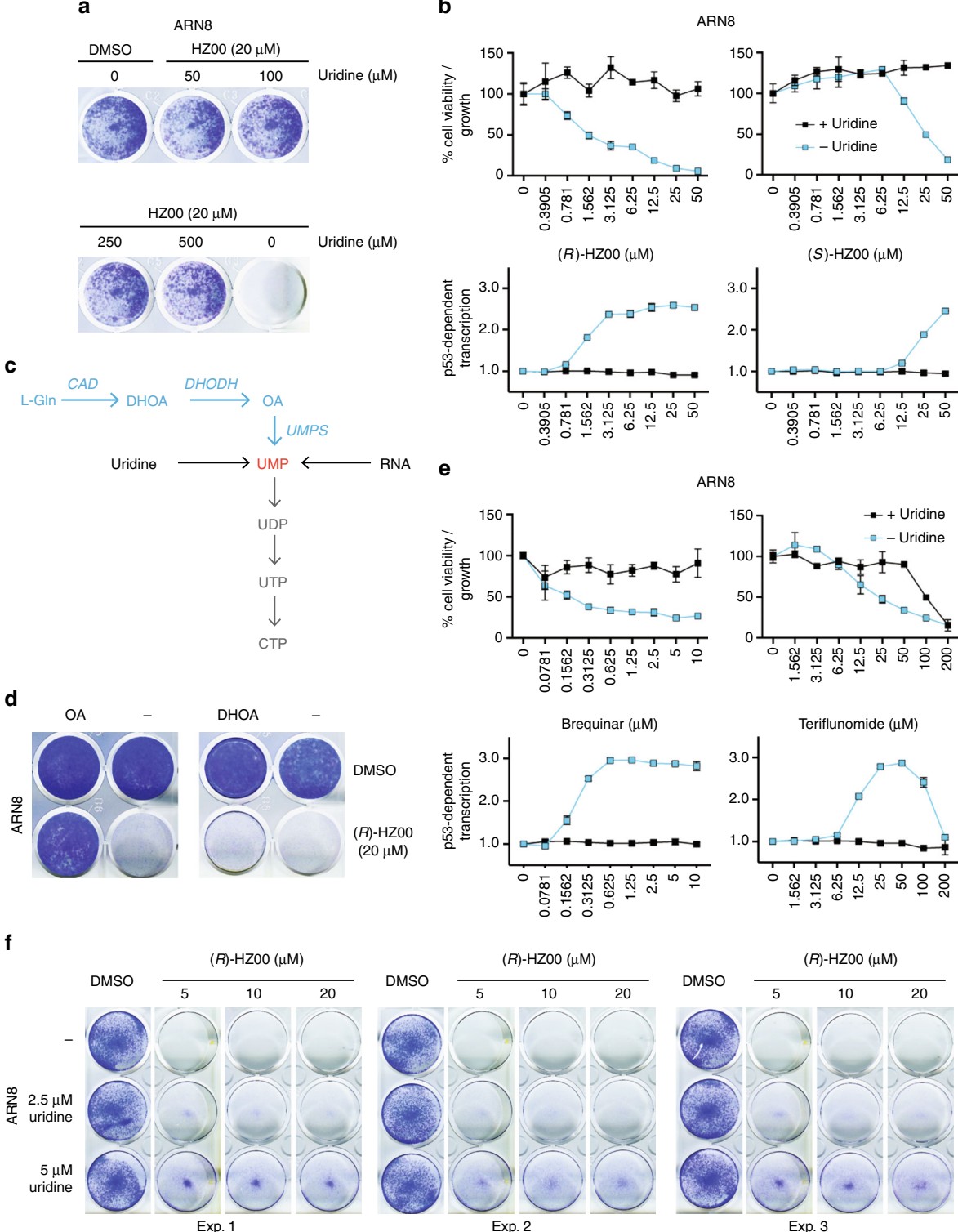

**Fig. 4** DHODH is a target of HZ00. **a** ARN8 cells were treated with HZ00 and the indicated amounts of uridine for 96 h followed by fixation with 50:50 methanol:acetone and subsequent staining with 7.5% Giemsa solution. **b** ARN8 cells were treated for 96 h with the indicated compounds in the presence and absence of uridine (100 μM) and analyzed using an SRB assay. Additionally, ARN8 cells were treated for 16 h and p53-dependent transcription was measured by CPRG assay. Values correspond to the average of three technical repeats ± SD. **c** Simplified description of the pyrimidine nucleotide de novo (blue) and salvage (black) pathways. **d** ARN8 cells were seeded in FBS supplemented DMEM with a change to serum replacement medium 24 h post seeding. Cells were treated for 6 days with (R)-HZ00 in the presence of 1.5 mM orotic acid (OA, Sigma #O2750) or 1.5 mM dihydroorotic acid (DHOA, Sigma #D7003) and stained with Giemsa as described in **a**. **e** ARN8 cells were treated as in **b** but using DHODH inhibitors brequinar and teriflunomide. Values correspond to the average of three technical repeats ± SD. **f** ARN8 cells were seeded in FBS supplemented DMEM with a change to serum replacement medium 24 h post seeding. Cells were then treated with (R)-HZ00 for 72 h in the presence of 2.5 or 5 μM uridine and stained with Giemsa as described in **a**

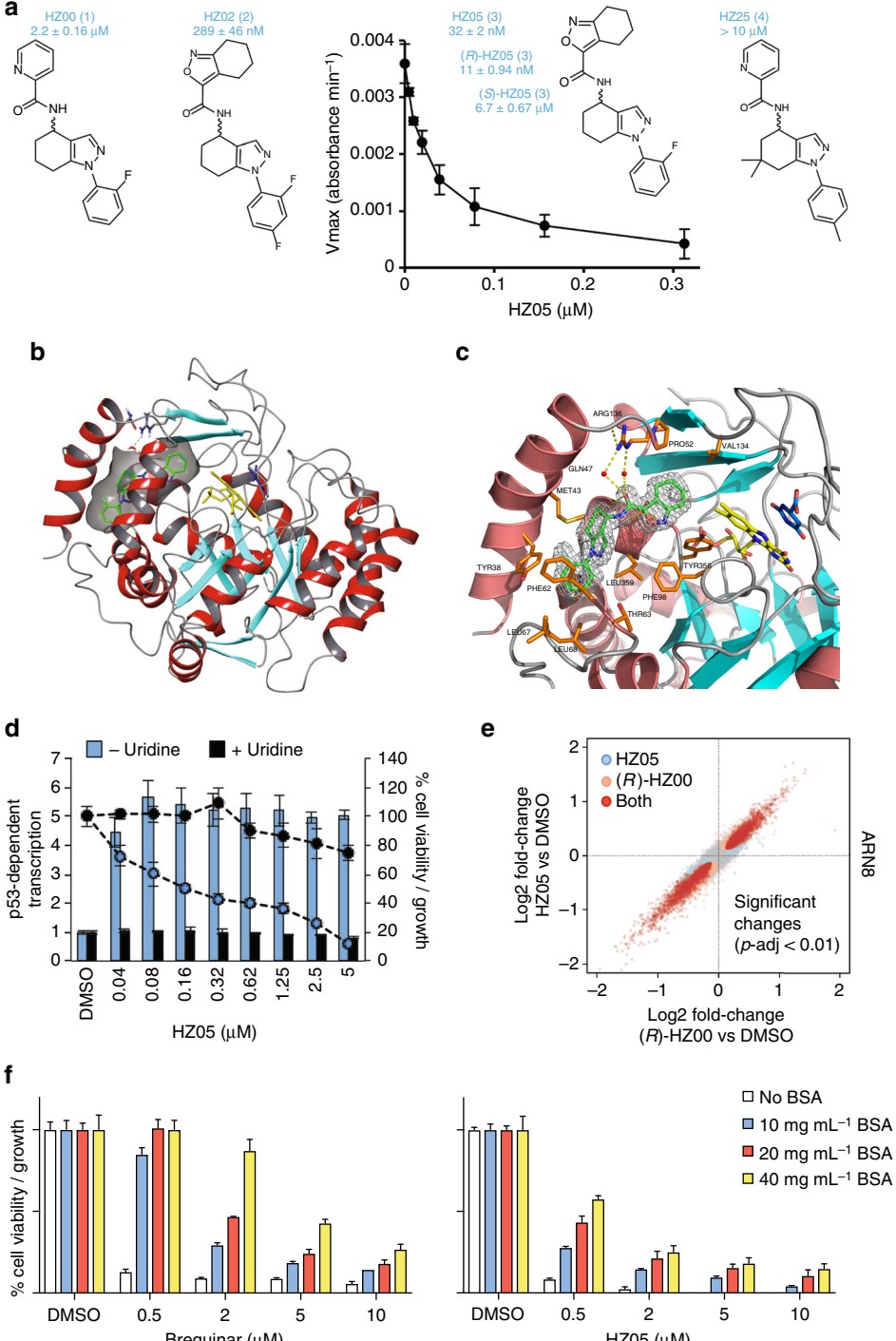

**Fig. 5** Characterization of the more potent analog HZ05. **a** Compound structures and $IC_{50}$ values obtained using a kinetic DHODH enzyme assay. Values correspond to the average of three independent repeats ± SD with three technical repeats each. **b** X-ray co-crystal structure of (*R*)-HZ05 (green carbons) with DHODH (6ET4). Protein helices are colored light red and the sheets in cyan. The carbons of the co-factor FMN are shown in yellow and the carbons of the reaction product orotate in blue. **c** (*R*)-HZ05, shown with green carbons, binds in the same pocket as other inhibitors such as brequinar. In this structure, the pocket is opened up as Gln47 (purple) is moved out, allowing (*R*)-HZ05 to make interactions with the Arg136 (orange) through two water molecules. Other amino acid residues within 4 Å from (*R*)-HZ05 are shown in orange. Ala55, His56, and Ala59 are located on helix number 3; however, the entire helix number 3 has been omitted for clarity. Also for clarity, Thr360 and Pro364 located below (*R*)-HZ05 are not labeled. The electron density of (*R*)-HZ05 is shown in light gray. Crystallography studies were performed as described in Supplementary Information. **d** p53-dependent transcription (bars) and viability/growth (lines) in ARN8 cell cultures in response to HZ05 in the absence and presence of 100 μM uridine. Values correspond to the average of three technical repeats ± SD. **e** Comparison between the effects of 20 μM (*R*)-HZ00 and 5 μM HZ05 on RNA levels (RNASeq experiment) in ARN8 cells treated with either compound for 5 h (Supplementary Data 1). **f** ARN8 cells were grown in DMEM supplemented with 5% FBS and treated with the indicated compounds for 72 h in the presence of increasing levels of bovine serum albumin (BSA). Following treatment, cells were processed for an SRB assay. Values correspond to the average of three technical repeats ± SD

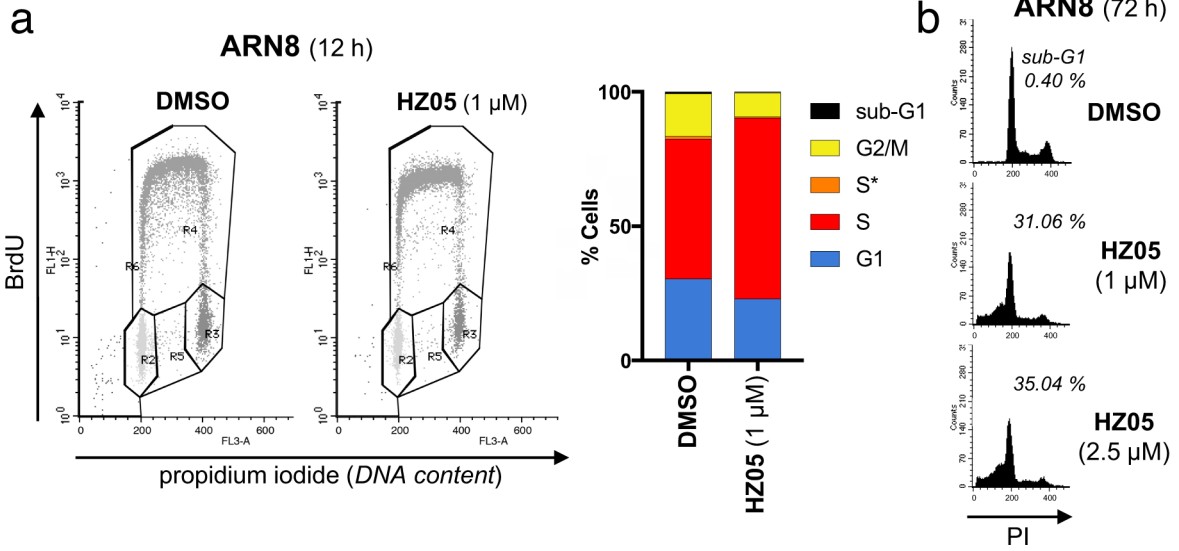

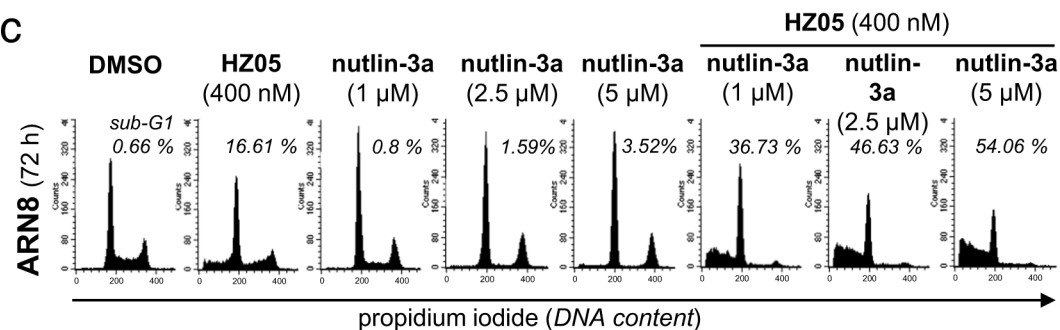

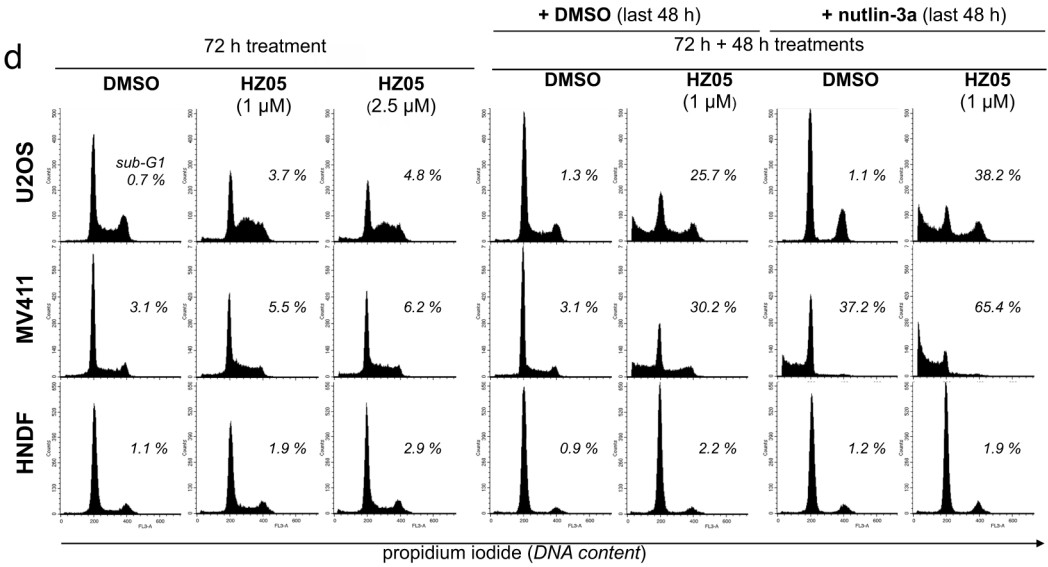

**Fig. 6** HZ05 leads to an enrichment of cancer cells in S-phase. **a** ARN8 cells were treated with vehicle (DMSO) or HZ05 for 12 h and analyzed using BrdU/PI flow cytometry. S* indicates cells with a DNA content between 2N and 4N that do not incorporate BrdU. **b** ARN8 cells treated for 72 h as indicated and then analyzed using PI flow cytometry. Numbers correspond to percentage sub-G1 cells. **c** ARN8 cells were treated with vehicle (DMSO), HZ05, nutlin-3a, or the combination for 72 h. Cells were analyzed by flow cytometry after staining with PI. Numbers indicate % of sub-G1 cells. **d** U2OS, MV411, and HNDF cells were treated with vehicle (DMSO) or HZ05 for 72 h (first three columns). In the next four columns, cells were treated for 72 h with either HZ05 or vehicle (DMSO) followed by re-addition of vehicle (DMSO) or HZ05 for an additional 48 h with or without 2 μM nutlin-3a. Cells were analyzed by flow cytometry following staining with PI. Numbers indicate % of sub-G1 cells

properties of the inactive analog HZ25 (**4**) (Fig. 5a) since the gem-dimethyl group in its central core appears to be too bulky to be accommodated by human DHODH.

Correlating with its ability to inhibit DHODH at low concentrations, HZ05 is also a potent inhibitor of ARN8 cell growth and an activator of p53. Furthermore, like previously shown for HZ00 (Fig. 4a, b), these effects were prevented by addition of excess uridine (Fig. 5d). Serving to confirm the similarity in their mechanism of action, (*R*)-HZ00 and HZ05 showed comparable results in an RNASeq analysis (Spearman correlation = 0.94 for genes with 50 reads or more) (Fig. 5e and Supplementary Data 1). For the list of p53-inducible mRNAs affected by HZ compounds see Supplementary Fig. 5a. We also report a list of mRNAs reduced by HZ compounds that are known to be downregulated upon activation of p53 (Supplementary Fig. 5b).

Following the successful identification of HZ05 as a DHODH inhibitor, we compared the effect of serum albumin concentration on the effectiveness of HZ05 or brequinar on cells. As has been described in previous studies, a carboxylic acid group is essential for the ability of brequinar to inhibit DHODH[10]; however, this same group is also responsible for the binding of compounds to serum albumin[14]. HZ05, like HZ00, lacks the carboxylic acid group present in brequinar. As shown in Fig. 5f, HZ05 is affected to a lesser degree than brequinar by increases in serum albumin concentration. This highlights that HZ compounds are part of a novel chemotype that may possess different pharmacokinetic properties to brequinar. It is also important to note that HZ05, much like (*R*)-HZ00, is able to ablate tumor cell growth in culture at physiologically relevant uridine levels (Supplementary Fig. 6a). In addition, brequinar has also shown itself to be effective at physiological uridine levels in vivo and was successful at ablating tumor cell growth[15].

Another similarity between HZ00 and HZ05 is that in spite of activating p53, both compounds lead to the accumulation of ARN8 cells in S-phase within 8 and 24 h of treatment (Figs. 3a and 6a). Also as described for HZ00 (Fig. 3a, b), the HZ05-induced accumulation of cells in S-phase was followed by an increase in the percentage of sub-G1 cells (Fig. 6b, c). Much like with HZ00, this increase in sub-G1 cells was enhanced by co-administration of nutlin-3a (Fig. 6c), the active enantiomer of nutlin-3[4]. We also saw that the co-treatment of ARN8 cells with HZ05 and nutlin-3a in vitro led to a strong synergistic cell kill (Supplementary Fig. 6b). Following the successful combination of HZ05 and nutlin-3a in cell culture, we carried out further xenograft studies. We found that as single agents, both nutlin-3a and HZ05 were ineffective at ablating xenograft tumor growth (Supplementary Fig. 7a). Despite the ineffective nature of the single treatments, combining nutlin-3a and HZ05 caused a statistically significant reduction in xenograft growth (Supplementary Fig. 7a). To better understand why HZ05 was not effective as a single agent, we examined it's in vivo pharmacokinetic properties. We found that the half-life of HZ05 was 2.5 h following subcutaneous administration (Supplementary Fig. 7b). Given that co-administration of nutlin-3a and HZ05 was markedly more effective than each compound as a single agent, we investigated whether the increased efficacy was due to drug–drug interactions leading to altered metabolism of nutlin-3a. However, in the case of both (*R*)-HZ00 co-administered with nutlin-3, and (*R*)-HZ05 with nutlin-3a, the half-life of nutlin-3 or nutlin-3a was not prolonged (Supplementary Figs. 7c and d). Additionally, we carried out xenografts using the p53-null H1299 cell line and dosed the mice using the same treatment regimen as in Supplementary Fig. 7a. There was no synergy between (*R*)-HZ05 and nutlin-3a in this p53-null xenograft (Supplementary Fig. 7e). This suggests that the inhibition of the degradation of

p53 by nutlin-3a is important for the synergy seen between (*R*)-HZ05 and nutlin-3a in wild-type p53 tumor xenografts.

**HZ compounds accumulate cells in S-phase with high levels of p53.** As we could confirm the ability of DHODH inhibitors to synergize with an inhibitor of p53 degradation in vivo with both HZ00 and HZ05, we further investigated the effects of this combination on other cell lines. Much like the ARN8 cell line, SigM5, MV411 and U2OS cells also demonstrated the ability of HZ compounds to accumulate cells in S-phase, followed by an increase in the proportion of sub-G1 cells (Supplementary Fig. 3). For both HZ00 and HZ05, these two consecutive events were faster in ARN8 cells than in SigM5, MV411, and U2OS cells. Also as in the case of ARN8 cells (Fig. 6c), adding nutlin-3a increased the proportion of dead cells in HZ05 treated U2OS and MV411 cultures (Fig. 6d). In contrast, there were only marginal increases in cell death in HNDF cultures treated with HZ05 alone or in combination with nutlin-3a (Fig. 6d).

U2OS cells (Fig. 6d), like ARN8 cells (Figs. 2c and 6c), do not die in response to nutlin-3 or nutlin-3a as single agents and instead accumulate in G1/G2. However, unlike the ARN8 cells, U2OS cultures are slower to accumulate in S-phase upon HZ treatment (Fig. 6a, d and Supplementary Figs. 3a and b). Furthermore, we observed that in order to see the promotion of cell death by nutlin-3a in U2OS cells, it was necessary to pre-treat with HZ05 and then add nutlin-3a (Supplementary Fig. 8). When HZ05 and nutlin-3a were added simultaneously to U2OS cells, the killing effect of HZ05 was ablated (Supplementary Fig. 8). We then tested whether cells in S-phase may have higher levels of p53 due to HZ05 treatment. Indeed, these S-phase cells possessed higher levels of p53 than vehicle treated controls (Fig. 7a).

Based on these results, we carried out a further experiment to see whether nutlin-3a, on its own, could cause cell death in U2OS osteosarcoma cells when they had been pretreated with HZ05. This hypothesis was supported by the results of the experiment shown in Fig. 7b, where we pretreated cells with HZ05 for 72 h, washed out the pre-treatment, and then added nutlin-3a as a single agent. This scheduling switched the effect of nutlin-3a from one that caused G1/G2 arrest to one that triggered cell death.

These results suggest that HZ compounds sensitize cells to inhibitors of mdm2 by accumulating them in S-phase with high levels of p53 (Fig. 7c).

## Discussion

In this study, we identify the enantiomer (*R*)-HZ00, as a cell active and selective inhibitor of DHODH. We also show through SAR and by examining the crystal structure of (*R*)-HZ05 bound to DHODH that it is possible to improve the medicinal chemistry properties of the HZ series for future studies and for lead compound development. This study also illustrates that a knowledge-based approach can lead to the identification of cellular targets for small molecules from phenotypic screens. This target deconvolution strategy is of particular relevance when the target is difficult to extract in conditions that retain a native conformation. In addition to being membrane bound, DHODH is protected by the outer mitochondrial membrane[16].

One implication of our work is that a high percentage of compounds identified in a simple phenotypic p53-activation screen are direct DHODH inhibitors. Several additional DHODH inhibitors with unrelated structures have been reported in the literature[10,17–22]. The wide range of chemotypes that can interact with and inhibit DHODH suggests that the enzyme may act as a target or sensor for xenobiotics, including the numerous and abundant metabolites in blood derived from the action of the microbiome[23]. Altogether, these results indicate that DHODH

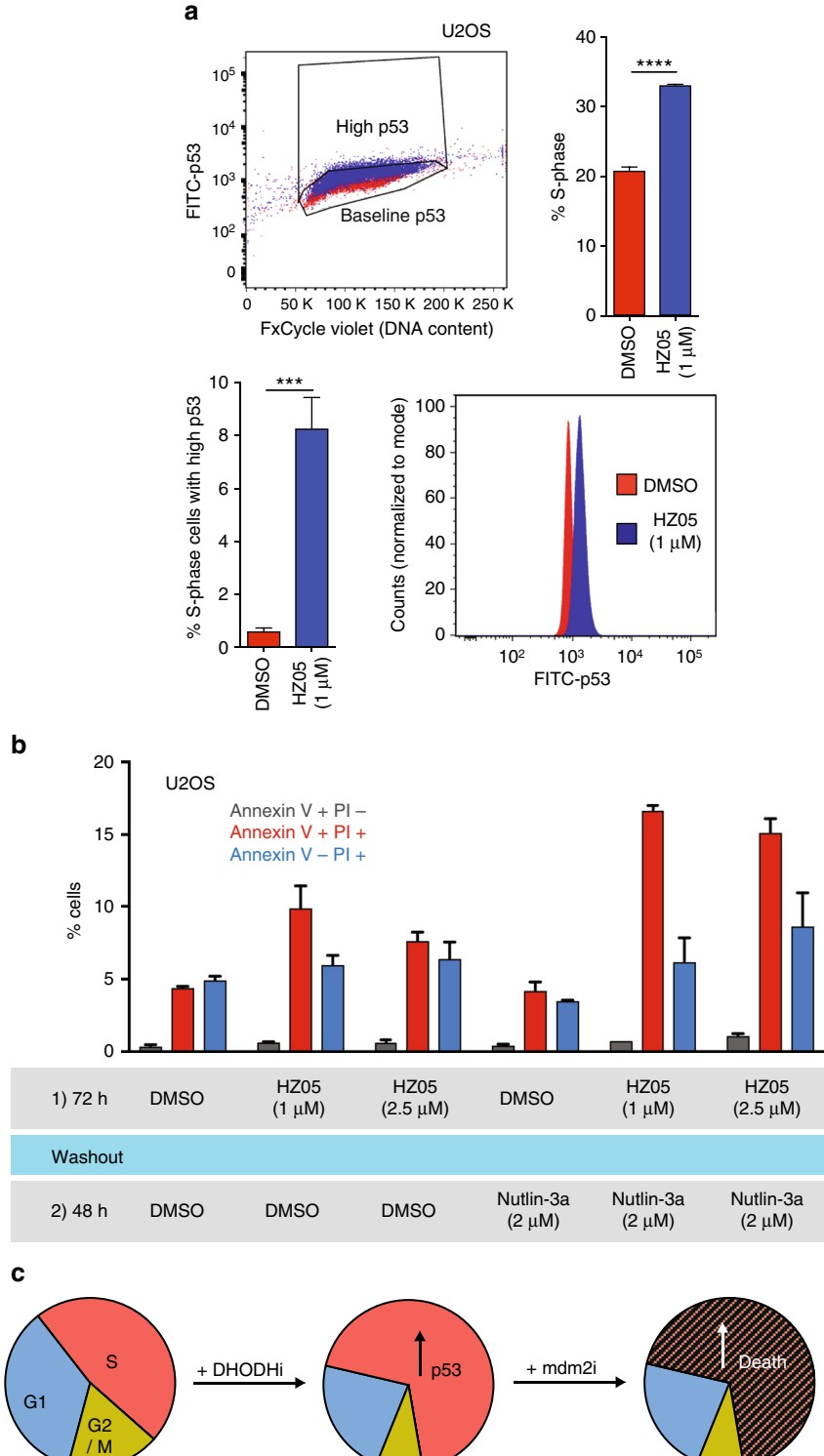

**Fig. 7** HZ05 increases p53 levels in S-phase cells. **a** U2OS cells were treated with DMSO (vehicle) or HZ05 (1 μM) for 72 h. The upper left panel is a representative dot plot showing the gating strategy for cells expressing high p53 levels (blue: HZ05 treated and red: DMSO treated). The upper right panel shows the percentage of cells in S-phase. The lower left panel shows the percentage of S-phase cells with p53 levels above baseline according to the gating shown in the dot plot. A representative histogram of the fluorescence intensity of FITC-p53 in S-phase cells is shown in the lower right panel. Bar graphs show the average of three independent experiments. Unpaired *t*-test was employed to assess statistical significance using GraphPad Prism version 7 (****$p$ < 0.0001 and ***$p$ < 0.001). **b** U2OS cells were pretreated with HZ05 for 72 h, washed with medium containing DMSO or nutlin-3a, and then provided with fresh medium containing DMSO or nutlin-3a. After an additional 48 h, Annexin V and PI staining was analyzed by flow cytometry. The average of two independent experiments ± SD is shown. **c** Hypothetical model describing the combination of DHODH inhibition with mdm2 antagonists. Inhibition of DHODH in tumor cells leads to an increase in the number of cells in S-phase that also possess elevated levels of p53. Inhibition of p53 degradation in the form of an mdm2 antagonist leads to death of cells in S-phase

should be considered as a potential target in phenotypically driven drug discovery projects.

The prodrug leflunomide and its active metabolite teriflunomide are clinically approved DHODH inhibitors used for treatment of rheumatoid arthritis and multiple sclerosis respectively[10]. However, the clinical use of leflunomide and teriflunomide is limited by off-target effects and exceptionally long persistence in the body[24]. Nevertheless, leflunomide reduces xenograft tumor growth[25–27]. With regards to its effects on p53, leflunomide, although itself inactive as a DHODH inhibitor, was shown to elevate p53 protein levels in HeLa cells, a cell line where p53 is modulated by viral oncoproteins rather than mdm2[28]. In a previous study, a very high concentration (100 μM) of a teriflunomide analog was shown to accumulate p53 in non-infected cells[29]. Here we demonstrate that DHODH inhibitors such as members of the HZ series and brequinar activate a p53-dependent reporter in cells, and do so at low concentrations. The specificity of these compounds for DHODH in cells is supported by the ablation of their effects on cell growth by excess uridine. Furthermore, it is important to note that DHODH inhibitors that interact with DHODH through a carboxylic acid group, also bind extensively to serum albumin, which alters their free unbound plasma concentration. This applies to compounds structurally related to brequinar. In contrast, the HZ series of compounds do not require a carboxylic acid group for their interaction with DHODH, thus providing an alternative to compounds related to brequinar.

We noted that HZ05 had a relatively short half-life in vivo. Following this observation, we have embarked on refining the structure to improve its pharmacokinetic properties to generate a promising lead compound. Nevertheless, despite the relatively short half-life of HZ05 in mice, we did confirm that as with HZ00, the combination of a HZ compound with a blocker of p53 degradation led to synergy in cell culture experiments and statistically significant reductions in tumor growth in vivo without evidence of toxicity.

Importantly, we also show that HZ compounds cause death in cancer cell cultures and that this is preceded by an accumulation of cells in S-phase, which is accompanied by the increased synthesis of p53. TP53 gene transcription is enhanced during the G1/S transition upon release from serum starvation[30], suggesting that the increase in p53 protein synthesis induced by HZ compounds could occur as a consequence of the accumulation of p53 mRNA in cells in S-phase. However, we could not see marked increases in p53 mRNA. Since there is no evidence of increased stability of p53 protein, a plausible hypothesis would be to consider that the increase in p53 protein is due to enhanced translation as reported for DNA damaging agents by Takagi et al.[31].

Another interesting feature noted in HZ treated cells is that p21 protein levels, but not mRNA levels, are relatively weakly induced compared to nutlin-3 (Fig. 1d). Furthermore, HZ compounds reduce the p21 levels induced by nutlin-3 treatment. On the one hand, this could contribute to accumulation of cells in S-phase, on the other hand it may also indicate a change in the amount of translation of p21 mRNA.

Whichever mechanisms hold true, we have demonstrated that HZ treated cultures possess more S-phase cells with higher p53 levels than untreated controls (Fig. 7a). Therefore, as depicted in the model in Fig. 7c, we propose that releasing p53 from the inhibitory effects of mdm2 during S-phase, especially when p53 is in excess, enhances p53's pro-apoptotic functions over its cell cycle inhibitory effect.

The discovery of new DHODH inhibitors, as well as a novel strategy to increase p53 activation and synergism with mdm2 inhibitors offers an exciting prospect to bring p53 therapy to fruition and may allow the cure of diseases like CML that retain resistance to elimination via a p53 sensitive stem cell population[2].

## Methods

**Cell culture**. ARN8 cells and T22 cells, stably expressing the p53 reporter RGCΔFos-LacZ were described previously[12,32–34]. H1299, U2OS, and MV411 cells were purchased from the ATCC and SigM5 were purchased from DSMZ. HCT116 cells were a kind gift from Professor B. Vogelstein (Johns Hopkins). HNDF cells were purchased from PromoCell. Cell lines were checked for mycoplasma contamination using the MycoAlert kit (Lonza LT07-318). HCT116 cells were grown in McCoy's 5A medium supplemented with 10% FBS and 100 U mL$^{-1}$ of pen/strep. SigM5 cells were grown in IMDM supplemented with 20% FBS and 100 U mL$^{-1}$ of pen/strep. All other cells were grown in DMEM and supplemented with 10% FBS and 100 U mL$^{-1}$ of pen/strep. For serum replacement studies, DMEM was supplemented with 1× serum replacement solution 3 (Sigma S2640). All cells not sourced from ATCC or DSMZ in the last year were checked using single tandem repeat analysis conducted by Public Health England. ARN8 cells were a 100% match to A375 cells, U2OS were a 100% match, H1299 were a 97% match and HCT116 cells used in Supplementary Fig. 2k were an 85% match. HCT116 cells used in Supplementary Figs. 1c and 4a were a match on 30 out of 32 alleles, but demonstrated multiple peaks at loci D7, D8, D13, D16, as well as FGA and vWA.

**Compound library screens for p53 activation (CPRG assay)**. A 20,000 compound library was purchased from ChemBridge consisting of 10,000 from the DIVERSet and 10,000 from the CombiSet libraries. ARN8 cells were treated with each compound at 10 μM for 18 h and β-galactosidase activity measured using the β-galactosidase CPRG substrate as previously described[12,32–34]. A total of 30,000 additional compounds from the ChemBridge DIVERSet that were previously screened in a T22 cell background[12] were re-screened in ARN8 cells at 5 μM. The ChemBridge codes for these compounds can be made available upon request. All chemical synthesis is detailed in Supplementary Information with NMR spectra and reaction schemes detailed in Supplementary Figs. 13–19.

**Western blotting and immunofluorescence**. Protein extracts were prepared in 1× LDS sample buffer (Invitrogen) with 100 mM DTT and separated and transferred using the Invitrogen western blotting system except in Supplementary Fig. 1c where the BioRad western blotting system was used. HRP-conjugated secondary antibodies were obtained from Dako (#P016102 and #P0211702) or Santa Cruz (#SC-2020). Immunofluorescence was performed by fixing cells in 4% paraformaldehyde freshly made in PBS for 10 min at 37 °C. Following fixation, cells were permeabilized in 0.15% Triton X-100 for 1–2 min at 37 °C followed by staining with the indicated antibodies. Images were taken using Olympus IX-71 microscope controlled by DeltaVision SoftWoRx. Image stacks were deconvolved, quick-projected and saved as tiff images to be processed using Adobe Photoshop. Antibodies to specific antigens are listed in Supplementary Table 8. All original films for blots in Fig. 1 are shown in Supplementary Figs. 9–12.

**p53 synthesis assay**. ARN8 cells were seeded at $2.5 \times 10^6$ cells per 10 cm dish. Next day, cells were treated for 5.5 h with DMSO, 20 μM HZ00 or 5 μM nutlin-3. After 5.5 h, medium was substituted for Met- and Cys-deprived DMEM, supplemented with 5 μM nutlin-3, 0.5% FBS and 4 mM L-glutamine for 20 min in the presence of the corresponding compounds. $^{35}$S-Met-Cys (50 μCi mL$^{-1}$) was added for 30 min. Cells were washed twice with PBS and scraped off in 800 μL per dish of 20 mM Tris, pH 7.5, 50 mM NaCl, 0.5% Triton X-114, 0.5% sodium deoxycholate, 0.5% SDS, 1 mM EDTA and sonicated. Samples were centrifuged at 16,000×g for 15 min at 4 °C. Volume of 30 μL of each supernatant was stored as input. The remaining supernatants were incubated with 5 μg anti-p53 DO-1 antibody and rotated overnight at 4 °C. Samples were transferred to Dynabeads Protein G (# D10004D, Life Technologies) and rotated for 30 min at RT. Samples were centrifuged briefly for 10 s and placed in front of a magnet. Pellets were washed twice with RIPA buffer (10 mM Tris-HCl, pH 7.4, 150 mM NaCl, 1% Triton X-114, 1% sodium deoxycholate, 0.1% SDS), once with high-salt buffer (10 mM Tris, pH 7.4, 2 M NaCl, 1% Triton X-114) and once with RIPA buffer. Pellets were resuspended in 3× LDS containing 100 mM DTT. Samples were separated by SDS-PAGE and the gels were incubated in enhancer solution (#6NE9741, PerkinElmer), dried and exposed to Carestream Kodak BioMax MR films at −80 °C.

**MTT and SRB growth/viability assays**. MTT assays were performed in 96 well plates. After treatments, medium was replaced with 200 μL of MTT solution (3 mg mL$^{-1}$ MTT (Sigma #M2128) in PBS diluted 1:5 in growth medium). Cells were incubated for 3 h after which medium was removed and substituted with 25 μL Sorensen's Glycine buffer (0.1 M glycine, 0.1 M NaCl, pH 10.5). Volume of 200 μL DMSO per well were added and plates were read at 570 nm.

SRB assays measuring protein content were performed in 96-well plates as previously described[35].

**Flow cytometry**. Cell cycle distribution analysis was performed on a Becton Dickinson FACScan flow cytometer or FACSCalibur (BD Biosciences) following

staining with propidium iodide (PI) or dual labeling with BrdU and PI[36]. These assays were analyzed using CellQuest Pro (BD Biosciences).

For the detection of apoptotic cells, the Annexin V-FITC Apoptosis detection kit (Ab#14085) was used with analysis conducted using FlowJo software V10.2.

For the detection of p53, cells were treated as indicated and fixed using Fix/Perm buffer solution (eBiosciences). Following fixation cells were stained with p53-FITC DO-7 (645804, BioLegend) in Permeabilization buffer (eBiosciences) according to manufacturer's instructions. Cells were resuspended in FACS buffer (2 mM EDTA, 0.5% BSA) containing 2 µg mL$^{-1}$ FxCycle Violet (Thermo Scientific). Stained cells were acquired on LSRII (BD Biosciences) and analyzed using FlowJo software V10.2.

**Endpoint and kinetic DHODH enzyme assays**. In Supplementary Tables 5 and 6, endpoint assays were performed with 10 nM recombinant human DHODH (#ENZ-642, Prospecbio). The reaction mixture consisted of 624.6 µM DL-dihydroorotic acid, 66.4 µM coenzyme $Q_{10}$, 66.4 µM 2,6-dichlorophenolindophenol sodium salt (DCIP) (all reagents purchased from Sigma-Aldrich) in enzyme buffer (50 mM Tris-HCl, pH 8.0, 0.1% Triton X-100, 150 mM KCl). Loss in absorbance by DCIP was measured at 595 nm after incubation at RT for 60 min.

In Fig. 5a enzyme assays were optimized and performed with 6 nM recombinant human DHODH prepared as described[13]. The reaction mixture for these kinetic assays consisted of 1 mM DL-dihydroorotic acid, 100 µM 3,4-dimethoxy-5-methyl-$p$-benzoquinone (#D9150, Sigma-Aldrich), and 100 µM DCIP in enzyme buffer. A stock solution of 20 mM DCIP was prepared in enzyme buffer and filtered through filter paper (20–25 µm pore size) just before use. Loss in absorbance by DCIP was measured at 595 nm at RT in a stepped time course (8 × 2 min, 8 × 3 min, 6 × 5 min). The observed decrease in absorbance over time was linear between 8 and 26 min. Therefore, for each concentration of inhibitor tested, a value for DHODH's $V_{max}$ was estimated by linear regression within this time frame. The IC$_{50}$ is defined as the concentration of inhibitor that gives $V_{max}$ ([I]) = $V_{max}$ (DMSO)/2.

**RNASeq**. Libraries were generated using the NEBNext Ultra Directional RNA Library Prep Kit for Illumina (New England Biolabs). Data are deposited at GEO with the accession code: GSE87577. In this site, compound M is (R)-HZ00 and compound A is HZ05 and used at 20 µM and 5 µM, respectively. Differential expression and gene ontology analysis are in Supplementary Data 1. Values correspond to the average of three biological repeats.

**Co-crystallization of (R)-HZ05 and human DHODH**. Co-crystals were prepared as described[13]. Details can be found in Supplementary Methods.

**Animal experiments**. All animal xenograft experiments were approved by the Norwegian Animal Research Authority and conducted according to The European Convention for the Protection of Vertebrates Used for Scientific Purposes. To determine the efficacy of (R)-HZ00 in combination with nutlin-3 in vivo, 20 z (5 per group as determined by power calculation assuming a fivefold difference and an expected $p$-value of <0.005 with a power of 85%) were injected subcutaneously in the left flank area with 5 × 10$^6$ ARN8 cells resuspended in 100 µL of PBS solution containing 12.5% Matrigel (BD Biosciences, Franklin Lakes, NJ, USA). The health status and weight of the mice were monitored daily and they were randomized in a non-blinded manner into 4 groups when tumor volumes reached 30–40 mm$^3$. The mice were either treated with vehicle solution (10% DMSO and 40% polyethylene glycol), (R)-HZ00 (at 150 mg kg$^{-1}$ q.d. in vehicle solution containing 10% DMSO and 40% polyethylene glycol 400 dissolved in sterile water), nutlin-3 (100 mg kg$^{-1}$ q.d. in a vehicle solution containing 2% hydroxypropyl cellulose and 0.5% Tween 80 dissolved in sterile water) or by a combination of the two drugs. The vehicle solution and (R)-HZ00 were given by intraperitoneal injection (i.p.) and nutlin-3 was administered orally (p.o.). Tumor volumes were measured every second day by a digital caliper using the following ellipsoid formula: Volume = π (length × width × height)/6.

To determine the efficacy of (R)-HZ05 in combination with nutlin-3a, 4 groups of NSG mice ($n$ = 8 per group) were subcutaneously injected in the flank region with a single 100 µL injection of 2 × 10$^6$ ARN8 cells in a solution of PBS:Matrigel (volume ratio 3:1). Treatment was initiated when tumors reached ~50 mm$^3$. The maximum tolerated dose for this combination of drugs was previously determined for the strain we used. The endpoint of the experiment was determined by the tumor size (<1000 mm$^3$) weight loss, and general condition of the mice according to The Norwegian Animal Research Authority.

**Data availability**. The atomic coordinates and structure factors are deposited in the Protein Data Bank with accession code 6ET4. GEO for the RNA seq. accession code: GSE87577. All data generated or analysed during this study are included in this article.

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

## Acknowledgements

The authors would like to acknowledge ChemAxon (www.chemaxon.com) for providing an academic license to their cheminformatics software and Sweden Contract In vivo Design AB for performing in vivo PK studies. We also thank Annika Lindqvist, Eliane Hesse, Melina Vallbracht, Levin Schulze, Amparo Martínez Pérez, and Antonio Ramírez Fernández for technical support, as well as EPRSC National Mass Spectrometry Service Centre (Swansea) for analytical data. Financial Support: M.J.G.W.L., C.J.D., I.M.M.v.L., G.P., T.M., S.D., M.C.C.S., A.P.-F., C.T., D.P.L., M.A.H., K.L., and S.L.: project grants from the Swedish Research Council, the Swedish Cancer Society and the Swedish Childhood Cancer Foundation. M.H. and J.C.: Cancer Research UK (C8/A6613). M.C., E.P., and W.C.E.: Wellcome Trust (073915). M.N. and B.V.: projects MEYS-NPS-LO1413 and GACR P206/12/G151. E.M.C., M.P., M.M.S., Z.F., and P.G.: Norwegian Cancer Society (182735, 732200) and Helse Vest (911884, 911789). R.B. and S.C.: NIH (R01 CA95684), the Leukemia and Lymphoma Society and the Waxman Foundation. N.J.W., A.R.H., A.C.A.d'H.: Cancer Research UK (C21383/A6950) and Engineering and Physical Sciences Research Council Doctoral Training Program. J.L. and Y.Z.: Cancer Research UK (C240/A15751). M.H. and B.W.: SARomics Biostructures AB. U.Y., K.F.: DDDP SciLife, Sweden. L.J., M.H., R.S., and A.-L.G.: CBCS, Sweden. VP: SciLife fellowship. AMT: Breast Cancer Research Scotland

## Author contributions

CPRG Assay for compound screen was conducted by J.C. and M.Hi. CPRG assays for the selection of HZ00 amongst active compounds from the screen was carried out by A.R.M., C.J.D. and S.L. The q-RT-PCR was conducted by J.C. and M.Hi. Western blotting was carried out by M.J.G.W.L., I.M.M.v.L., C.J.D., A.P.F., M.C.C.S., M.N. and B.V. Flow Cytometry (propidium iodide) was done by M.J.G.W.L., I.M.M.v.L., C.J.D., S.C., G.P., T.M. and A.P.-F. Flow Cytometry (BrdU/PI) was conducted by M.J.G.W.L., C.J.D. and G.P. Flow Cytometry using Annexin/PI was done by MJGWL. Flow Cytometry examining p53/DNA content was carried out and analyzed by M.J.G.W.L. and S.K.S. SRB. Viability Assays were done by I.M.M.v.L., S.D. and A.P.F. and the MTT. Viability Assay by C.J.D. The CPRG p53 transcriptional activity assay (excluding the screen) were carried out by I.M.M.v.L., C.J.D., S.D. and A.P.F. Clonogenic assays were done by M.J.G.W.L., A.P.F. and C.T. The HDM2/HDMX binding assay (ELISA) was carried out by Y.Z. and J.L. Immunofluorescence was carried out by E.M.P., M.C. and W.C.E. The DHODH enzyme activity assay was done by I.M.M.v.L., G.P. and C.T. All RNASeq was done by G.P. and V.P. All [35]S Radiolabelling was done by I.M.M.v.L. Protein Crystallography was carried out by A.-L.G., M.Håk., and B.W. Chemical Synthesis of compounds and their characterization was done by A.H., M.Har., L.J., U.Y., A.-L.G., K.F., A.C.A.d.H. and N.J.W. In vivo xenograft experiments were carried out by M.P., M.M.S., P.G., Z.F., E.M. and A.M.T. In vitro pharmacokinetic properties were carried out by R.S. and K.F. while the in vivo pharmacokinetic analysis was conducted by A.S. Analysis (e.g., statistical analysis, bio-statistics, computational analysis) and interpretation of data was done by M.J.G.W.L., S.K.S., M.P., M.M.S. A.-L.G., M.Håk., B.W., V.P., M.A.H., D.P.L., E.M. and S.L. Supporting data on effects of compounds on various cell types was done by M.J.G.W.L., I.M.M.v.L., C.J.D., G.P., T.M., S.C., M.F., K.L., M.N., B.V. and R.B. Writing, of the manuscript was carried out by M.J.G.W.L., C.J.D., N.J.W. and S.L. Review, and/or revision of the manuscript was conducted by M.J.G.W.L., I.M.M.v.L., C.J.D., A.H., L.J., U.Y., A.-L.G., M.Håk., B.W., N.J.W., M.A.H., D.P.L., E.M. and S.L. Administrative, technical, or material support (i.e., reporting or organizing data, constructing databases) was organized by G.P., M.Håk., and V.P. All studies were supervised by S.L.

## Additional information

**Competing interests:** S.L. has filed a patent application on the use of HZ compounds for cancer treatment, which was sent for publication on 4 May 2017. The remaining authors declare no competing interest.

Marcus J.G.W. Ladds[1,2], Ingeborg M.M. van Leeuwen[1], Catherine J. Drummond[1], Su Chu[3], Alan R. Healy[4], Gergana Popova[1], Andrés Pastor Fernández[1], Tanzina Mollick[1,2], Suhas Darekar[1,2], Saikiran K. Sedimbi[1,2], Marta Nekulova[1,5], Marijke C.C. Sachweh[1], Johanna Campbell[6], Maureen Higgins[6], Chloe Tuck[1], Mihaela Popa[7], Mireia Mayoral Safont[7], Pascal Gelebart[7], Zinayida Fandalyuk[7], Alastair M. Thompson[8], Richard Svensson[9], Anna-Lena Gustavsson[10], Lars Johansson[10], Katarina Färnegårdh[11], Ulrika Yngve[12], Aljona Saleh[12], Martin Haraldsson[11], Agathe C.A. D'Hollander[4], Marcela Franco[1], Yan Zhao[13], Maria Håkansson[14], Björn Walse[14], Karin Larsson[1], Emma M. Peat[15], Vicent Pelechano[2], John Lunec[13], Borivoj Vojtesek[5], Mar Carmena[15],

William C. Earnshaw[15], Anna R. McCarthy[1], Nicholas J. Westwood [4], Marie Arsenian-Henriksson[1], David P. Lane[1,2], Ravi Bhatia[3], Emmet McCormack[7,16] & Sonia Laín[1,2]

[1]Department of Microbiology, Tumor and Cell Biology (MTC), Karolinska Institutet, SE-171 77 Stockholm, Sweden. [2]SciLifeLab, Department of Microbiology, Tumor and Cell Biology (MTC), Karolinska Institutet, Tomtebodavägen 23, SE-171 21 Stockholm, Sweden. [3]Division of Hematology and Oncology, Comprehensive Cancer Center, 1720 2nd Avenue South, NP2540, Birmingham, AL 35294-3300, USA. [4]School of Chemistry and Biomedical Sciences Research Complex, University of St. Andrews and EaStCHEM, St. Andrews, Fife, Scotland KY16 9ST, UK. [5]RECAMO, Masaryk Memorial Cancer Institute, Zluty Kopec 7, 65653 Brno, Czech Republic. [6]Centre for Oncology and Molecular Medicine, University of Dundee, Ninewells Hospital and Medical School, Dundee, Tayside DD1 9SY, UK. [7]Centre for Cancer Biomarkers, CCBIO, Department of Clinical Science, Hematology Section, University of Bergen, 5021 Bergen, Norway. [8]Department of Breast Surgical Oncology, MD Anderson Cancer Center, Holcombe Boulevard, Houston 77030, USA. [9]Department of Pharmacy, Uppsala University Drug Optimization and Pharmaceutical Profiling Platform (UDOPP), Department of Pharmacy, Uppsala University, SE-752 37 Uppsala, Sweden. [10]Chemical Biology Consortium Sweden, Science for Life Laboratory, Division of Translational Medicine and Chemical Biology, Department of Medical Biochemistry and Biophysics, Karolinska Institutet, SE-171 21 Stockholm, Sweden. [11]Drug Discovery and Development Platform, Science for Life Laboratory, Tomtebodavägen 23, SE-171 21 Solna, Sweden. [12]Department of Medicinal Chemistry, Science for Life Laboratories, Uppsala University, SE-751 23 Uppsala, Sweden. [13]Newcastle Cancer Centre, Northern Institute for Cancer Research, Newcastle University, Newcastle NE1 7RU, UK. [14]SARomics Biostructures, Medicon Village, SE-223 81 Lund, Sweden. [15]The Wellcome Trust Centre for Cell Biology, Institute of Cell Biology, University of Edinburgh, Edinburgh EH9 3JR, UK. [16]Department of Medicine, Haematology Section, Haukeland University Hospital, Bergen, Norway. These authors contributed equally: Marcus J.G. W. Ladds, Ingeborg M.M. van Leeuwen, and Catherine J. Drummond. Deceased: Anna R. McCarthy

