## [Peer Review File · Nature Communications]

Reviewers' comments:

Reviewer #1 (Remarks to the Author):

This manuscript reports on a multidisciplinary effort to discover p53 activators. This screening being based on the indirect effect of nutlin-3 an mdm2 inhibitor which this releases p53 from its binding this negative regulator. As for quite a few recent screening programs seeking a rather diverse set of biological effects, DHODH inhibitors stood out in this screening. From one of these hits (HZ00; wisely chosen in view of the many hits of very little interests which are listed in the supplementary tables 5 and 6; by the way, an actual depiction of their structures would be welcome), a short structure activity relationship investigation using commercially available analogues led to HZ05 a much stronger inhibitor of this enzyme. Moreover, an X-ray based structure of the binding mode of this compound to DHODH.

Concerning this aspect of the report, few additional references should be incorporated somewhere in the discussion in order to illustrate the fact that many human DHODH inhibitors (including lipophilic compounds devoid of a carboxylic function) were recently found in the course of quite a variety of phenotypic-based assays (cf.: Proc. Natl. Acad. Sci. U.S.A. 2011, 108, 5777; PLoS Pathog. 2013, 9, e1003678, J. Med. Chem. 2015, 58, 5579; ACS Med. Chem. Lett. 2016, 7, 1112; Cell 2016, 167, 171; J. Med. Chem. 2015, 58, 1123)

At this stage, the manuscript proceeds to describe an extensive number of biological assessment of the effects of HZ00 and HZ05 which are very weak in cells (despite plenty of data concerning the stage of cell arrest, the purpose of them being to establish a causative link between these compounds an p53 concentration in cellulo) but these weak effects are somehow rescued by the co-addition of nutlin-3 the mdm2 inhibitor mentioned above. Then, HZ05 as well as the much weaker HZ00 (why bother, what kind of ethical committee authorized this and by the way at what stupendous dose was is done?) were also assessed in vivo in a xenograft model in the course of which the half-life of HZ05 was found to be of 2.5 hours. The remarkable paradox of this study (and many other previously reported in vivo DHODH inhibition investigations) is that since in cellulo, the addition of uridine "ablate" the effect of DHODH inhibitors, the large blood concentration of uridine in vivo should also "ablate" any DHODH-based inhibition effects of this type of compounds. This observation is leading to the conclusion that the in vivo effects reported here are either only due to the administration of nutlin-3 (but the control experiment using only nutlin-3 seems to disagree with this) or to an "off-target" action of HZ05. One of the many possible "off-target" would be the inhibition of the catabolism of nutlin-3 by HZ05 which would increase the concentration and thus the effect of this mdm2 inhibitor. Indeed, both compounds are aryl-rich lipophilic substances probably subject to extensive liver-based decomposition which could be somehow "saturated" by HZ05.

In conclusion this referee while being impressed by the amount of work described (especially the medchem part) does not believe that the results and conclusion drawn from this work are of a sufficient validity and value to be published in their present form.

Reviewer #2 (Remarks to the Author):

The manuscript by Ladds and coworkers describes the identification of 20 compounds from a library of 20,000 that are able to increase transcription from a p53-dependent reporter. One compound named HZ00 is studied in detail. HZ00 alone and in combination with the Mdm2 inhibitor nutlin-3 is able to increase expression of p53 and the reporter. HZ00 also reduces proliferation of treated cells. It is the (R)-enantiomer of HZ00 that is active. It leads to an increase in the sub-G1 cell population in one cell type. HZ00 is found to be an inhibitor of dihydroorotate dehydrogenase (DHODH). X-ray structures confirm this mode of inhibition. Based on these initial observations, more inductors of p53 and

inhibitors of DHODH are identified. These compounds lead to an increase of p53 that is accompanied by an accumulation of cells in S phase and a strong increase of the sub-G1 population. The inhibitors are also tested in xenograft tumor models.

Comments:

This is an important study to identify new compounds able to activate p53. The manuscript covers many aspects relevant for such a search with a wide spectrum of methods. There are several points that require discussion and possibly experimentation.

What is the connection from inhibiting dihydroorotate dehydrogenase to increased levels and activation of p53?

The authors discuss that the accumulation of p53 at G1/S may be a result of increased transcription. Is the level of p53 mRNA elevated under the experimental setting during that cell cycle phase? Or is it p53 protein stability that increases, as this hypothesis may be supported by elevated effect once nutlin-3 is included in the assays?

There is some discussion on p53 protein levels in relation to reporter activity and increase in p21 levels. The relative level of p21 expression should be regarded as a measure for p53 transcriptional activity. It appears that p53 levels do not necessarily correlate with p21 levels and thus with p53's transcriptional activity. Is it a modification of p53 such as phosphorylation or some other covalent switch that is influenced indirectly by the new class of effector compounds?

It is not clear why in the ARN8 cells p21 decreases in the presence of HZ00 from a high level upon addition of nutlin-3 (Fig. 1d).

The authors describe that they do not observe an upregulation of cdc6 (Suppl. Fig. 1i-j). This is not surprising because cdc6 is a p53-DREAM pathway target (Sci Rep. 2017 Jun 1;7(1):2603). Thus, with this pathway intact in a cell, activation of p53 will result in a decrease of cdc6 levels.

p53 target genes are reported for their change upon (R)-HZ00 or HZ05 treatment (Suppl. Fig. 4). As the data acquired will also yield p53 targets which are downregulated, the list should be complemented by adding repressed genes as well.

Kurt Engeland

Reviewers' comments and Answers:

Reviewer #1 (Remarks to the Author):

1. *This manuscript reports on a multidisciplinary effort to discover p53 activators. This screening being based on the indirect effect of nutlin-3 an mdm2 inhibitor which this releases p53 from its binding this negative regulator. As for quite a few recent screening programs seeking a rather diverse set of biological effects, DHODH inhibitors stood out in this screening. From one of these hits (HZ00; wisely chosen in view of the many hits of very little interests which are listed in the supplementary tables 5 and 6; by the way, an actual depiction of their structures would be welcome), a short structure activity relationship investigation using commercially available analogues led to HZ05 a much stronger inhibitor of this enzyme. Moreover, an X-ray based structure of the binding mode of this compound to DHODH.*

Response:

We thank the reviewer for this comment. We are pleased to read that the reviewer thinks that the HZ00 compound was wisely chosen and we have added representative structures for the strongest compounds from each chemotype (**New Supplementary Table 7**). Whether the other compounds listed in Tables 5 and 6 are also interesting cannot be answered without a detailed analysis of all of them. Our main point in showing this result was that it is remarkable that so many DHODH inhibitors were identified using a simple p53-based phenotypic screen, a point that we still consider valid and of interest for any drug screening approach.

2. *Concerning this aspect of the report, few additional references should be incorporated somewhere in the discussion in order to illustrate the fact that many human DHODH inhibitors (including lipophilic compounds devoid of a carboxylic function) were recently found in the course of quite a variety of phenotypic-based assays (cf.: Proc. Natl. Acad. Sci. U.S.A. 2011, 108, 5777; PLoS Pathog. 2013, 9, e1003678, J. Med. Chem. 2015, 58, 5579; ACS Med. Chem. Lett. 2016, 7, 1112; Cell 2016, 167, 171; J. Med. Chem. 2015, 58, 1123)*

Response:

We thank the reviewer for the diligence in finding these references. These references have now been added to the manuscript. These papers further cement the concept presented in our manuscript that DHODH inhibitors can exist in various chemical spaces, making DHODH a rather an important target to be screened against in future drug discovery projects.

3. *At this stage, the manuscript proceeds to describe an extensive number of biological assessment of the effects of HZ00 and HZ05 which are very weak in cells (despite plenty of data concerning the stage of cell arrest, the purpose of them being to establish a causative link between these compounds an p53 concentration in cellulo) but these weak effects are somehow rescued by the co-addition of nutlin-3 the mdm2 inhibitor mentioned above.*

Response:

HZ00 was the primary hit from the screen (Figure 1b). HZ05, a commercially available HZ00 analog, was subsequently purchased following target deconvolution of HZ00. HZ05 was validated as a DHODH inhibitor using an enzymatic assay and phenotypic readouts for DHODH inhibition. At this point, HZ05 was identified to be a much more potent compound in the enzymatic assay and on cultured cells. HZ05 was able to activate p53-dependent transcription and inhibit DHODH enzymatic activity at concentrations below 40 nM, as well as reduce cell viability, or growth within the 160 nM range. Accordingly, the DHODH active enantiomer of HZ05, (R)-HZ05, is even more active in cells. This is in contrast to the very widely used mdm2 inhibitor nutlin-3 which is only active in these assays at μM

concentrations but has of course led to extensive pharmaceutical efforts from ROCHE, AMGEN, MERCK SANOFI and others, underlining the importance of tool compounds in pharmaceutical development.

The effect of HZ00 and HZ05 was potentiated (not rescued) upon addition of nutlin-3/nutlin-3a as shown in Figures 2c, 6c and 6d. We have added new data to further demonstrate this synergy (**New Fig. 2d and New Supplementary Fig. 6b**). We can rescue the effect of HZ00 and HZ05 on cells but this is upon the administration of uridine or orotic acid as shown in Figures 4a and 4d, a feature that supports specificity for DHODH as the target for HZ compounds.

4. *Then, HZ05 as well as the much weaker HZ00 (why bother, what kind of ethical committee authorized this and by the way at what stupendous dose was is done?) were also assessed in vivo in a xenograft model in the course of which the half-life of HZ05 was found to be of 2.5 hours.*

Response:

HZ00, as the first compound discovered in our phenotypic screen for activators of p53 transcriptional activity, was first tested *in vivo* following the observations of efficacy *in vitro* as well as selectivity against tumor cells (Figures 1e, 2b and 2c) as well as for possessing an interesting enantiomer specific activity as detailed in Figure 2b. Following a successful assessment of efficacy *in vivo* (Figure 2d), target elucidation of HZ00 was conducted and a screen of further HZ00 analogs against the molecular target of HZ00 was carried out. From this screen, a number of more potent analogs were discovered eventually culminating in the co-crystallization of the compound most able to inhibit DHODH activity *in vitro*, HZ05. Of great satisfaction was the finding that the co-crystal selected the *R*-isomer from the enantiomeric mix, underlining at the structural level the great specificity of this compound series. To identify a molecular fingerprint of the HZ series we conducted an RNAseq and carried out a similarity analysis. We found a very high degree of correlation between HZ00 and HZ05 (Figure 5e).

Following these analyses we conducted a second *in vivo* study to assess efficacy of HZ05 as a single agent and in combination with nutlin-3a. Whilst we acknowledge that the half-life of HZ05 is not optimal, between HZ00 and HZ05 we have discovered compounds that are enantiomer-specific and that demonstrate cell-type specificity as well as possessing the ability to potentiate nutlin-3 both *in vitro* and *in vivo*.

These molecules have now served as a molecular scaffold for further development into compounds that not only demonstrate potency (as HZ05 has) but also demonstrate *in vivo* activity as single agents due to more favorable pharmacokinetics (as HZ00 has). A thorough SAR paper with 90 HZ analogues is in preparation.

5. *The remarkable paradox of this study (and many other previously reported in vivo DHODH inhibition investigations) is that since in cellulo, the addition of uridine “ablate” the effect of DHODH inhibitors, the large blood concentration of uridine in vivo should also “ablate” any DHODH-based inhibition effects of this type of compounds.*

Response:

We carried out experiments using literature values of uridine in plasma (between 2.5-4.9 μM in human plasma and between 1.2-3.2 μM in murine plasma), see review by Traut TW, Mol Cell Biochem. Nov 9 1994; 140(1) 1-22) to determine the effectiveness of HZ compounds on cells under such conditions. We have found, that though activity is inhibited to a degree, the HZ compounds, much like the gold standard literature compound brequinar, were still able to ablate the growth of tumor cells (these data have now been introduced in the new version of

the manuscript, see **New Figure 4f and New Supplementary Figure 6a**). 100 μ M concentrations of uridine (more than 20 fold the serum concentration of uridine) are used simply as a route to establish on target specificity in our cell based assays. As mentioned above, we clearly demonstrate that the compounds are active in cell culture medium supplemented with 2.5 or 5 μ M uridine thus solving the referee's reasonable concern. In addition to our work, brequinar has also shown itself to be effective at physiological uridine levels *in vivo* and was successful at ablating tumor cell growth (Dexter et al. Cancer Res 1985; 45 5563-8 and Sykes et al. Cell 2016; 167(15) 171-186).

6. *This observation is leading to the conclusion that the *in vivo* effects reported here are either only due to the administration of nutlin-3 (but the control experiment using only nutlin-3 seems to disagree with this) or to an "off-target" action of HZ05.*

One of the many possible "off-target" would be the inhibition of the catabolism of nutlin-3 by HZ05 which would increase the concentration and thus the effect of this mdm2 inhibitor. Indeed, both compounds are aryl-rich lipophilic substances probably subject to extensive liver-based decomposition which could be somehow "saturated" by HZ05.

Response:

This is a highly pertinent question. However, we do not believe this is due to an off-target effect of HZ compounds based upon the experiments showing that the effects of HZ compounds on cells (even when used at very high concentrations) are rescued by supplementation with 100 μ M uridine (Figure 4 and Figure 5d) and the fact that HZ compounds in combination with nutlins are highly synergistic in killing ARN8 melanoma cells (Figure 2c and e, **New Figure 2d**, Figure 6c, **New Supplementary Figure 6a**, Supplementary Figure 7a). Additional support for specificity of these compounds can be found in Supplemental Tables 1-3.

Regardless of target specificity in cell culture, and here we fully agree with the reviewer, it is always important to test for drug interactions *in vivo* as highlighted in this comment. We thank the reviewer for this observation and agree that this could be a confounding factor in the study. We have, therefore, now conducted an *in vivo* pharmacokinetic study of single and co-administered HZ00 and nutlin-3, as well as HZ05 and nutlin-3a using the same routes of administration and dosing as detailed in the xenograft studies. We found that there was no increase in the plasma levels of either nutlin-3 or nutlin-3a upon co-administration of HZ compounds. We have now added this data to the supplementary information of the manuscript (**New Supplementary Figure 7c-d**). Again, these experiments do not support the thesis that HZ compounds stabilize nutlins.

Because it could be a concern that nutlin's off-target p53-independent effects could be enhanced by HZ compounds, we also have extra data using p53 null cells (see below). We are happy to add this data to the manuscript if the reviewer finds it pertinent.

Effects of HZ05 alone and in combination.

In vivo activity of (R)-HZ05 alone and in combination with nutlin-3 on H1299 p53-null xenograft tumors ($n \geq 6$). In the tumor volume graph (*left*), (**) on day 4 indicates a statistical difference of Control to HZ05 and to nutlin-3a; (*) on day 7 indicates a difference between Control and nutlin-3a; and (**) on day 12 indicates a statistical difference between Control, Combination and nutlin-3a. In the normalized tumor volume graph (*right*), (***) on day 4 indicates differences between Control and the other three. There is no significant difference as determined by Student's t-test between the combination and any of the single treatments. *** $p < 0.001$, ** $p < 0.01$ and * $p < 0.05$

7. *In conclusion this referee while being impressed by the amount of work described (especially the medchem part) does not believe that the results and conclusion drawn from this work are of a sufficient validity and value to be published in their present form.*

Response:

We thank the reviewer for the insightful comments on potential “off target” interfering effect of HZ compounds on nutlin stability *in vivo* and the comment regarding uridine concentration in plasma. These are very reasonable questions and we hope that our answers now satisfy these important concerns.

Reviewer #2 (Remarks to the Author):

1. *This is an important study to identify new compounds able to activate p53. The manuscript covers many aspects relevant for such a search with a wide spectrum of methods. There are several points that require discussion and possibly experimentation.*

Response:

We thank the reviewer for his kind words and hope that our answers are sufficient clarification to their salient points. We appreciate his clear expertise related to p53 function and regulation.

2. *What is the connection from inhibiting dihydroorotate dehydrogenase to increased levels and activation of p53?*

The authors discuss that the accumulation of p53 at G1/S may be a result of increased transcription. Is the level of p53 mRNA elevated under the experimental setting during that cell cycle phase? Or is it p53 protein stability that increases, as this hypothesis may be supported by elevated effect once nutlin-3 is included in the assays?

Response:

This is an excellent question that deserves a lot of attention as the regulation of p53 synthesis is less studied than the regulation of its degradation. Furthermore, understanding how depletion of ribonucleotides can lead to an increase in the synthesis of p53 is very intriguing and important in our opinion.

In order to explain the increase in p53 protein synthesis observed by adding DHODH inhibitors, increases in p53 gene transcription, transcript splicing, mRNA stability and/or its translation need to be detected. Therefore, following the suggestion by the reviewer we have analysed these issues using RNAseq data (**New Supplementary File** ENSG00000141510) and PCR data. We have seen at both early (6 h) and late (18 h) timepoints that p53 mRNA levels are not significantly altered (**New Supplementary Figure 1b and d**).

We believe that the RNASeq data we have generated will be extremely useful for any researcher interested in modulation of p53 synthesis and may also help to identify transcripts that are most vulnerable to impaired transcription or most susceptible to degradation. We did not include the RNASeq complete data set in the first version of the manuscript but we have now done so as we do think that many readers, not only from the p53 field, will be interested (**New Supplementary File**). To our knowledge, this sort of data is only available after long-term treatments with inhibitors of ribonucleotide synthesis, and therefore more likely to reflect indirect effects. Our data is derived from a 5 hour treatment.

Further to studies of mRNA levels, we have also examined the effect of DHODH inhibition on p53 stability. Unlike the reference compound, nutlin-3, which increases p53 stability by binding its negative regulator, mdm2, DHODH inhibitors did not increase p53 stability in a cyclohexamide experiment detailed in Supplementary Figure 2b. We believe this is a strong indication that the increased amount of p53 protein and increased p53 transcriptional activity is through another mechanism beyond stabilization of p53. We are gathering further evidence that the effect is due to a rise in translation triggered by an alteration of several factors (proteins and lincRNAs) that act at different time points. We feel, however, that this is beyond the scope of a this paper and is best suited to a mechanistic study of DHODH inhibitors in general (manuscript in preparation).

We have modified the discussion according to the recent observations brought to light by this comment.

3. *There is some discussion on p53 protein levels in relation to reporter activity and increase in p21 levels. The relative level of p21 expression should be regarded as a measure for p53 transcriptional activity. It appears that p53 levels do not necessarily correlate with p21 levels and thus with p53's transcriptional activity. Is it a modification of p53 such as phosphorylation or some other covalent switch that is influenced indirectly by the new class of effector compounds?*

Response:

p21 mRNA and protein expression levels can increase due to p53-dependent or to p53-independent effects. For example, it is well known that other small molecules such as the HDAC inhibitor trichostatin A can lead to increased p21 mRNA and protein expression in HCT116 p53-deficient cells as shown by us (Sachweh et al. Cell Death Dis. 2013 Mar 7;4:e533).

HZ compounds increase in p21 mRNA according to qPCR (**New Supplementary Figure 1b**). Indeed, p21 mRNA levels rise to an extent comparable to that observed with nutlin-3, and they also rise in the 5h RNASeq experiment (See CDKN1A in **New Supplementary Figure 5a**). As mentioned below (see answer to point 4 by Reviewer #2), it is the p21 protein

levels that are increased more weakly with HZ compounds than with nutlin-3 (Figure 1c in the manuscript).

In order to ascertain that DHODH inhibition leads to p53-dependent p21 expression, in this new version of the manuscript we provide data using isogenic HCT116 cells showing that p21 mRNA and protein increase in HCT116 p53-wild type cells but do not rise in HCT116 p53-deficient cells (**New Supplementary Figure 1c**). For completeness, we show the same experiment for mdm2 and add that the p53 reporter construct expressing β -gal cannot be activated by HZ00 in a p53 null background. We thank the reviewer for this important question.

This data together with the observed increase in multiple p53-induced mRNAs (RNAseq experiment – see **new Supplementary File**), strongly suggest that the HZ compounds do activate p53-dependent transcription.

With regards to potential p53 posttranslational modifications in response to DHODH inhibitors we have checked for serine-15 p53 phosphorylation in response to HZ00 but did not detect any increase (Supplementary Figure 2e and f). Preliminary results from a phosphoproteomic study conducted for another manuscript indicate no rise in other phosphorylations of p53. We agree that it would be interesting to pursue this avenue of reasoning further and are doing so for a future manuscript utilising proteomic approaches, however, we feel that currently it is beyond the scope of this paper.

4. It is not clear why in the ARN8 cells p21 decreases in the presence of HZ00 from a high level upon addition of nutlin-3 (Fig. 1d).

Response:

p21 mRNA levels rise to an extent comparable to that observed with nutlin-3, and they also rise in the 5h RNASeq experiment (See CDKN1A in **New Supplementary Figure 5a**). Interestingly, it is the p21 protein levels that are increased more weakly with HZ compounds than with nutlin-3 (Figure 1c). Thus, upon DHODH inhibition, an event that reduces p21 mRNA translation and/or an event that reduces the stability of p21 protein may take place. Alternatively, it is possible that nutlin-3 could directly or indirectly increase p21 translation or its stability better than DHODH inhibitors.

As the reviewer mentions, it is interesting that the combination of nutlin-3 with HZ compounds leads to a weaker induction of p21 protein levels than observed with nutlin-3 on its own. This indeed is very intriguing though, as mentioned above, this weakening by the compound combination is not evident at the p21 mRNA level.

The hypothesis we are currently testing to explain the HZ-induced reduction in p21 protein levels in the presence of nutlin-3 involves a positive modulator of p21 mRNA translation as we have observed that the levels of this RNA binding protein drop upon DHODH treatment. Therefore, we suggest that whilst p21 mRNA levels are raised upon DHODH inhibition, p21 translation is partially impaired. In this way, the strong positive effect of nutlin-3 on p21 protein levels may be weakened by co-incubation with HZ compounds. However, this mechanism needs to be verified and we cannot expect that this translation modulator is the only factor involved in modulating p21. The RNAseq data suggests that other proteins and non-coding RNAs are likely to be involved (**New Supplementary File**). Preliminary experiments testing whether p21 stability is altered upon DHODH inhibition suggest that this is not the case, but these need further confirmation.

5. The authors describe that they do not observe an upregulation of *cdc6* (Suppl. Fig. 1i-j). This is not surprising because *cdc6* is a p53-DREAM pathway target (Sci Rep. 2017 Jun 1;7(1):2603). Thus, with this pathway intact in a cell, activation of p53 will result in a decrease of *cdc6* levels.

Response:

We thank the reviewer for noticing this as we did not know that *cdc6* could be downregulated through the p53-DREAM pathway. We agree with the reviewer and in the new version mention the fact that *cdc6* could be downregulated as a consequence of activation of the p53-DREAM target pathway. As shown in the figure below below the downregulation of *cdc6* by HZ00 is indeed p21-dependent which is interesting considering that p21 is necessary for the p53-DREAM pathway to inhibit transcription of genes. However, we believe that whilst this finding is interesting, addition of this data would obfuscate the general message of our study.

Figure Legend: HCT116 p21 wt or null cells (Vogelstein lab) were treated for 18 h with indicated compounds and were analyzed by western blotting.

6. p53 target genes are reported for their change upon (R)-HZ00 or HZ05 treatment (Suppl. Fig. 4). As the data acquired will also yield p53 targets which are downregulated, the list should be complemented by adding repressed genes as well.

Response:

We have analyzed our RNASeq data for DHODH inhibitor induced changes on the expression of 211 genes known to be affected by the DREAM complex. It is remarkable that so many of the p53-p21-DREAM-CDE/CHR signaling pathway dependent genes (72 of 211, padj ≤ 0.0003) are downregulated after a short 5 hour exposure to HZ compounds (**New Supplemental Fig. 5b**).

Here we would like to mention that we have also updated our list of HZ compound induced genes whose expression is known to be upregulated by p53 in human cells according to the Espinosa lab (Allen et al., eLife 2014;3:e02200). To this list we have added mir34HG (**New Supplementary Figure 5a**).

We have also analyzed whether mir34a dependent mRNAs (Lal et al., PLoS Genet. 2011 Nov;7(11):e1002363) are affected by HZ compounds. However, we have not detected clear decreases in these mRNAs upon a 5h HZ treatment. This could be because our treatment time is too short or because mir34HG is not being processed upon HZ compound treatment. In any case, these results are far less clear than the observations we now report for the p53-DREAM complex modulated mRNAs (**New Supplementary Figure 5b**).

We hope that the inclusion of new data and our responses are helpful and clear. We are in the process of preparing a manuscript much along the line of two of the questions from the reviewer. We are happy to provide access to the data and manuscript if the answers are insufficient. We do, however, believe that further elucidation of these mechanisms within this paper is beyond the current scope of the study.

Reviewers' comments:

Reviewer #1:

This reviewer was not available to review this revision. Reviewer #2 assessed your responses to this reviewer and found that all the points had been adequately addressed. Regarding point 6, he/she does agree that it is a good idea to include the data using p53 null cells in the final manuscript.

Reviewer #2 (Remarks to the Author):

5.

Response:

We thank the reviewer for noticing this as we did not know that cdc6 could be downregulated through the p53-DREAM pathway. We agree with the reviewer and in the new version mention the fact that cdc6 could be downregulated as a consequence of activation of the p53-DREAM target pathway. As shown in the figure below below the downregulation of cdc6 by HZ00 is indeed p21-dependent which is interesting considering that p21 is necessary for the p53-DREAM pathway to inhibit transcription of genes. However, we believe that whilst this finding is interesting, addition of this data would obfuscate the general message of our study.

Revision Reviewer Comment:

I disagree. Since the authors have obtained the data on p21-dependence of cdc6, there is nothing really confusing for the reader to include them in Supplements.

Revision Minor Comments:

The introduction starts out with imatinib. However, except for a general mentioning of CML, the drug does not appear anywhere else later in the manuscript, questioning whether this is a good start to introduce the reader to the results from the paper.

Was there any prior selection for the particular 20,000 and later with the 30,000 compound libraries?

The p53 reporter RGCΔFos-LacZ is not described in the reference (Lain et al. 2008). A central reagent like this reporter must be described in sufficient detail in order to evaluate the validity of the screening results.

Reviewers' comments and Answers:

Reviewer #1 (Remarks to the Author):

1. Reviewer Comment:

This reviewer was not available to review this revision. Reviewer #2 assessed your responses to this reviewer and found that all the points had been adequately addressed. Regarding point 6, he/she does agree that it is a good idea to include the data using p53 null cells in the final manuscript.

Response:

We thank reviewer #2 for stepping in and assessing our comments to reviewer #1 and are happy that they feel we have addressed their concerns sufficiently. We have added the **New Supplementary Figure 7e** as suggested by the reviewer examining the effect of nutlin-3a on p53-null H1299 xenografts in combination with HZ05 and the resultant lack of synergy.

Reviewer #2 (Remarks to the Author):

1. Our first response to Reviewer #2:

We thank the reviewer for noticing this as we did not know that cdc6 could be downregulated through the p53-DREAM pathway. We agree with the reviewer and in the new version mention the fact that cdc6 could be downregulated as a consequence of activation of the p53-DREAM target pathway. As shown in the figure below below the downregulation of cdc6 by HZ00 is indeed p21-dependent which is interesting considering that p21 is necessary for the p53-DREAM pathway to inhibit transcription of genes. However, we believe that whilst this finding is interesting, addition of this data would obfuscate the general message of our study.

Revision Reviewer Comment:

I disagree. Since the authors have obtained the data on p21-dependence of cdc6, there is nothing really confusing for the reader to include them in Supplements.

Response:

We thank the reviewer for their comment and have added the blots in the HCT116 p53 +/- and -/- cells in **New Supplementary Figure 1c**.

2. Reviewer Comment:

The introduction starts out with imatinib. However, except for a general mentioning of CML, the drug does not appear anywhere else later in the manuscript, questioning whether this is a good start to introduce the reader to the results from the paper.

Response:

This is a salient point. We have now amended the introduction detailing why imatinib was an important case study for targeted therapeutics and provided a better link through to the second paragraph.

3. Reviewer Comment:

Was there any prior selection for the particular 20,000 and later with the 30,000 compound libraries?

Response:

We have specified the exact nature of the compound selections used in the text. The first set of 30 000 compounds was not selected by ourselves, but comes from the Chembridge selected

DIVERSet, which has been put together to provide a broad spectrum of pharmacophores for screening purposes. Within the 20 000 compounds screen, 10 000 compounds were from the second DIVERSet, which is selected by Chembridge, and the second set were selected by ourselves from the Chembridge CombiSet Library. HZ00 came from the second DIVERSet screen. The entire DIVERSet contains 50 000 compounds, and we have tested 40 000 of these.

4. Reviewer Comment:

The p53 reporter RGCΔFos-LacZ is not described in the reference (Lain et al. 2008). A central reagent like this reporter must be described in sufficient detail in order to evaluate the validity of the screening results.

Response:

This was well spotted by the reviewer. As well as highlighting the past study where we used this very screen (Lain et al. 2008), we have also added the original paper where this construct was produced (Frebourg et al. 1992), the production and use of T22 cell line (Lu et al. 1996) and the production and use of ARN8 cells (Blaydes et al. 1998).